# On the mechanics of inhaled bronchial transmission of pathogenic microdroplets generated from the upper respiratory tract, with implications for downwind infection onset

Saikat Basu *

Department of Mechanical Engineering, South Dakota State University, Brookings, South Dakota, United States of America

* Saikat.Basu@sdstate.edu

## Abstract

Could the microdroplets formed by viscoelastic fragmentation of mucosal liquids within the upper respiratory tract (URT) explain the brisk onset of deep lung infection following initial URT infections? Generally, particulates, inhaled through the nostrils and therefore navigating the intricate topography of the anterior nasal cavity, can efficiently reach the lower airway only if they are small enough, typically $\lesssim 5\mu$m. However, the fate of larger particulates, many exceeding 5-$\mu$m in diameter, that are sheared from the initial infection sites along the intra-URT mucosa during inhalation remains unresolved. These particulates originate primarily from the nasopharynx, oropharynx, and the laryngeal chamber containing the vocal folds. To investigate, this study employs a computed tomography-based three-dimensional anatomical airway reconstruction, isolating the tract from the larynx and mapping the tracheal cavity through to the third generation of the tracheobronchial tree; constituent transport across the distal bronchial outlets is also recorded to assess deep lung penetration. Within the defined geometry, airflow simulations are conducted with the Large Eddy Simulation scheme to replicate relaxed inhalation at 15 L/min flow rate. Against the ambient air flux, numerical experiments are performed to monitor the downwind transport of particulates (aerosols/droplets) with diameters $1 - 30$ $\mu$m, bearing physical properties akin to aerosolized mucus with embedded virions. The full-scale numerical transmission trends, representatively validated against a small set of published experimental data, are consistent with findings from our reduced-order mathematical model that conceptualizes the influence of intra-airway vortex instabilities on local particle transport through point vortex idealization in an anatomy-guided two-dimensional potential flow domain. The results collectively demonstrate a markedly elevated lower airway penetration by URT-derived particulates, even by those as large as 10 and 15 $\mu$m. The high viral load, often exceeding the pathogen-specific infectious dose, carried by such droplets into the bronchial spaces of the sample

**Data availability statement:** All data are fully available without restriction, via figshare (see https://doi.org/10.6084/m9.figshare.29954564).

**Funding:** National Science Foundation CAREER Award (Grant Number 2339001; Fluid Dynamics program with Dr. Ron Joslin as program manager). The funders had no role in study design, data collection and analysis, decision to publish, or preparation of the manuscript.

**Competing interests:** The author has declared that no competing interests exist.

airway, provides a plausible mechanistic explanation for the accelerated seeding of secondary lung infection.

## Introduction

When inhaled air sweeps past the mucociliary coating of the upper respiratory tract (URT), interfacial interactions lead to localized viscoelastic stretching and breakup of mucosal layers resulting in the formation and release of microdroplets [1–3], which could then be pushed downwind by the airflow streamlines. Prominent instability effects, when the viscoelastic layer (mucus) resting on a viscous fluid film (serous fluid) is exposed to the incoming airflow, are to be noted in this context; e.g., see [4] for the instability phenomena and [5] on modeling approaches for the viscoelastic behavior. The mechanism modulating the microdroplet formation is analogous to numerous observations from the reverse process: exhalation [6–9]. The expiratory transport regimes, for intense respiratory events [10] and even during silent breathing [11,12], often involve mucus fragmentation and subsequent emission of liquid particulates spanning a wide range of length scales. From that paradigm, if we pivot our attention to inhaled (i.e., *into* the airway) transport, some immediate questions that come forth concern the fate of the intra-URT particulates generated during inhalation and their probable relevance in progressive disease transmission, especially to the deep lungs. On that note, the present study, through numerical experiments performed in a three-dimensional anatomical airway reconstruction and with simulation-informed reduced-order analytical validation in an anatomy-inspired two-dimensional channel, attempts to answer the following main queries:

$Q_1$. If inhaled from outside and consequently navigating the complex anterior nasal topography, the dominant inertial motion as well as gravitational impaction for particulates larger than 5 $\mu$m may, in general, prevent their penetration to the lower airway [13,14]. However, is this also true for particulates of similar sizes shed through mucus separation at the back of the nasal passage from the URT sub-sites like the nasopharynx, oropharynx, and from around the vocal folds during inhalation? Hypothetically, with such particulates still being airborne at the larynx (see Fig 1), the relatively straight structural shape of the downwind tract up to the tracheal base may facilitate their penetration into the lower airway.

$Q_2$. What is the viral load transmitted to the lower airway by particulates originating from the infected intra-URT tissue surfaces? Could this mechanism systemically explain the rapid onset of lung infection following the initial infection and emergence of symptoms at URT sub-sites, such as the nasopharynx? First proposed (to the author's knowledge) in [15] using an idealized one-dimensional *trumpet* model with appropriate modifications to approximate the dichotomous structure of the lower respiratory tract (LRT), the veracity of such a mechanism in a three-dimensional anatomical domain can dispel the caveat concerning time-scale inconsistencies whereby attributing the rapid downwind progression of infection solely to tissue-level replication of the invading pathogen could be a stretch.

This study addresses $Q_1$ through full-scale numerical tracking of inhaled constituents inside a representative anatomically accurate airway geometry with a supporting reduced-order anatomy-guided mathematical model of the system. The anatomical test geometry extends up to the third generation of bronchial branching, with cavity outlets leading into finer respiratory bronchioles; see Fig 1. To generate a summative assessment of bronchial deposition alongside deep lung penetration of URT-derived particulates, this study extracts and combines: (a) the simulated bronchial deposition percentages up to generation 3; and (b) the percentage of particulates progressing further downwind through the distal bronchial outlets into the deeper respiratory bronchioles. To note, the term *deep lungs* refers to the lower regions of the respiratory system where gas exchange primarily occurs. This domain includes the bronchioles, alveolar ducts, and alveoli [16]. A detailed analysis of, say, the alveolar deposition trends in the deeper regions of the lungs lies beyond the scope of the present methodology. Using physiological modeling in a test respiratory domain, this paper exclusively assesses the lower airway penetration potential for URT-derived pathogenic microdroplets and how the mechanism could be correlated to secondary infection onset. However, for supplementary reading, comprehensive bronchial deposition profiles for (mostly pharmaceutical) particles administered nasally and/or orally from outside can be found in several excellent studies; e.g., see [17–23].

Next for $Q_2$, the fluid dynamics findings on particulate penetration are integrated with virological parameters, such as the sputum viral concentration [25] for a representative pathogen (SARS-CoV-2), and the projected viral load transmitted to the bronchial pathways is compared against the verified infectious dose of the pathogen [26,27]. The resulting translational analysis helps assess the proclivity for lower airway infection that is driven by inhalation of aerosolized intra-URT

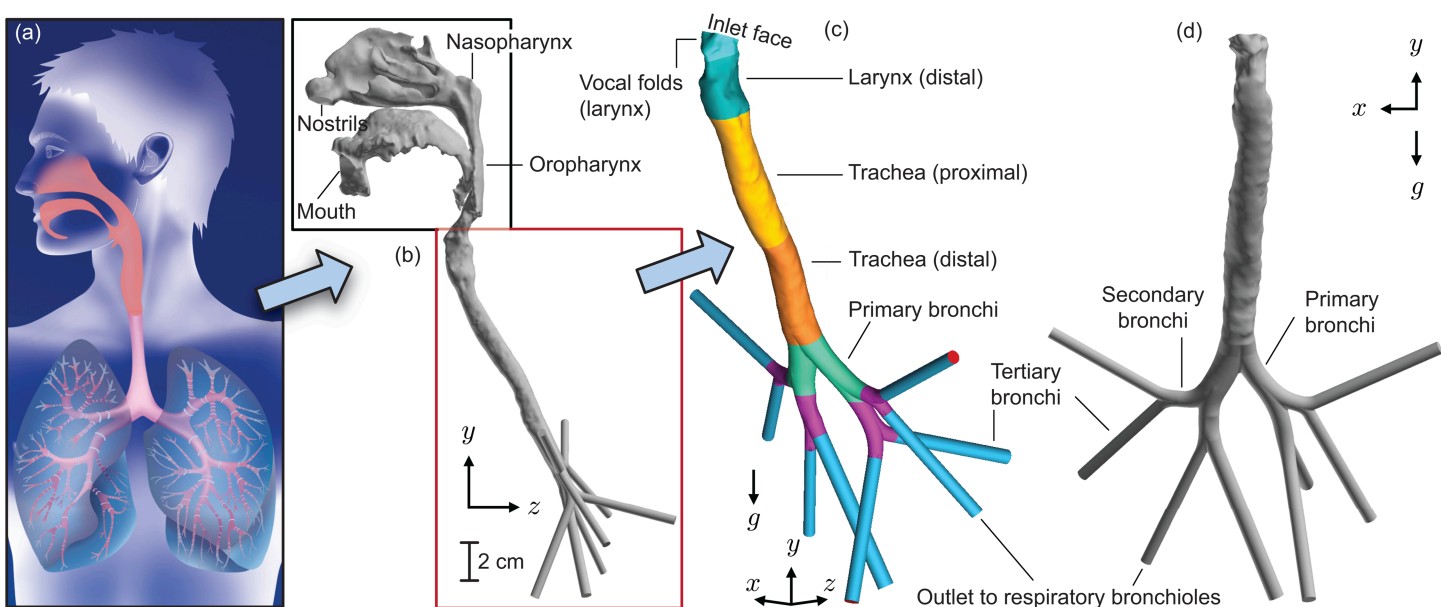

**Fig 1. Defining the physiological domain.** (a) A demonstrative cartoon of the human respiratory system, encompassing the upper respiratory tract (URT; comprising nasal cavity, pharynx, larynx), the mouth, and the lower respiratory tract (LRT; comprising regions downwind from trachea onward), extending till the deep lungs. The visual is adopted with a perpetual license agreement from the Getty Images®. (b) Sample computed tomography (CT) imaging-based reconstruction of an adult human airway. It serves as a three-dimensional anatomically realistic equivalent of the cartoon in panel (a), with regions included till generation 3 (tertiary bronchi) of the tracheobronchial tree. For confirmation, note that there are 8 distal outlets (see panels (b)-(d)), implying $2^{G_n} = 8$, where $G_n = 3$ is the generation number, considering two-way bifurcation for each bronchial tube at every transition [24]. The domain within the red box is isolated for the numerical experiments on inhaled downwind transport of microdroplets generated from the URT. The isolated region is additionally shown in panel (c) for an isometric view with anatomical demarcations and in panel (d) for the front coronal view. The symbol $g$ signifies the gravity direction in the numerical simulations and the subsequent analytical framework, with $x$, $y$, and $z$ defining the spatial orientation of the test cavity. Panel (b) additionally highlights the geometric length scale.

mucosal remnants. It is worth noting that this in silico approach, rooted in the underlying respiratory flow physics as discussed here, is agnostic to any virus specificity and potentially extensible to other respiratory pathogens by integrating the fluid mechanics outcomes of inhaled transport with appropriate virological and immunological data. In that context, this work can contribute toward advancing an emerging niche sub-field that brings together classical fluid dynamics and pathogen biology [26,28–33]. For select recent relevant studies on deep lung kinetics and pathophysiology, the reader may refer to [15,34–37]; additionally, one may peruse [38,39] for engaging accounts of airway flow physics, its potential impact on lung function, and the pertinent open problems.

## Materials and methods

### Numerical modeling of inhaled downwind transport of URT-derived microdroplets

As is the case for inhaled transport of microdroplets (also referred to as particulates, or equivalently simply at times as particles in this exposition) through the respiratory cavity, in numerical simulations involving the dispersion of small particles under dilute conditions—the common approach assumes one-way coupling. This reduction implies that while the airflow continuum carries the particulates, the impact of such particulates on the underlying flow regimes could be disregarded. Consequently, the ambient airflow field is initially resolved, and the flow outcomes are subsequently employed to numerically solve the relevant particle equations of motion.

In this study, we have used Large Eddy Simulation (LES) scheme with dynamic subgrid-scale kinetic energy transport model to numerically replicate the inhaled air flux through an anatomically realistic airway reconstruction. To model the inhaled particle transport therein, we have applied the Lagrangian approach which is more suitable (compared to the Eulerian methods) for the dilute suspension of relatively large particles for which inertia could often be dominant in determining the spatial trajectories and eventual deposition spots. The simulated airflow field is coupled with the Lagrangian particle transport analysis to derive the intra-airway deposition and penetration trends.

**Anatomical geometry reconstruction, spatial discretization, and mesh sensitivity analysis.** This analysis utilizes existing, de-identified, high-resolution, medical-grade computed tomography (CT) imaging, with slices acquired at coronal depth increments of $\approx 0.4$ mm. The CT scanning had exposed the subject to an additional radiation dose of 84 mrem, which is roughly equivalent to the natural background radiation that most individuals receive over four months. Note that the retrospective use of the anonymized data (resourced from the Department of Otolaryngology/Head and Neck Surgery in the School of Medicine at the University of North Carolina Chapel Hill) for computational analysis was approved under an exempt status by the Institutional Review Board (IRB) at South Dakota State University. The corresponding IRB determination number is: IRB-2206003-EXM [40]. From the medical imaging, this study has first reconstructed a complete three-dimensional adult respiratory airway [41]; see Fig 1a-b. The extraction warranted a radiodensity thresholding between −1024 to −300 Hounsfield units [42,43] to capture the airspace from the CT slices. The segmentation of the DICOM (Digital Imaging and Communications in Medicine) format CT scans was performed on the image processing software Mimics Research 18.0 (Materialise, Plymouth, Michigan). While the scanned data guided the shapes of the primary bronchi (with the right main bronchus bearing a wider diameter than the left), the subsequent generations of secondary and tertiary bronchi were digitally engineered into the test domain following Weibel's model of repeated bifurcations of the human respiratory tree [44]. The final geometry was quality-checked by a respiratory care specialist at the author's institution (see Acknowledgments) for anatomical realism.

Aiming to address $Q_1$ and $Q_2$ (pitched in the Introduction), the anatomical reconstruction was then digitally redacted to focus on the space mapping the vocal fold region of the larynx (space with highest concentration of liquid particulates formed from intra-URT mucosal fragmentation) along with the distal laryngeal chamber, the trachea, and the lower airway extending till generation 3 of the tracheobronchial tree, incorporating the primary, secondary, and tertiary bronchi, followed by the respiratory bronchiolar outlets as entry to the deeper recesses of the lungs. See Fig 1c-d for the redacted test geometry with the described regions marked out. The above redaction operation, along with the subsequent mesh

development (described next), was carried out by importing the CT-derived geometry (as a stereolithography file) to ICEM CFD 2024 R1 (ANSYS Inc., Canonsburg, Pennsylvania).

To prepare the domain for numerical simulations, a grid refinement analysis was conducted with the test cavity being spatially segregated into 0.5, 1.0, 1.5, 2.0, 2.5, and 3.0 million graded, unstructured, tetrahedral elements, along with four layers of pentahedral cells (with 0.025-mm height for each cell and an aspect ratio of 1.1) extruded at the airway cavity walls [45] to resolve the near-wall particulate dynamics; see Fig 2a-h. The variables assessed in the sensitivity study included the resistance $\mathcal{R}$ (in Pa.min/L) to the simulated inhaled airflow, calculated as $|\Delta P|/Q$, where $|\Delta P|$ in Pa represents the inlet-to-outlet static pressure gradient and $Q$ denotes the volumetric flux in L/min; the area-weighted average airflow velocity magnitude $V_o$ (in m/s) at the outlet surfaces of the geometry; and the transmission efficiency $\eta$ (in %) within the bronchial pathways for representative particulate sizes of 1, 5, 10, and 20 $\mu$m. The deposition and penetration efficiency $\eta$ quantifies the cumulative bronchial transmission trend for each particulate size—summing the deposition rates at the primary, secondary, and tertiary bronchi, together with the penetration rate through the distal bronchiolar outlets into the deeper lung regions (see Fig 1c-d). The fluctuation trends of the tracked variables are shown in Fig 2i-k. Specifically, the simulation results yield:

$$
\begin{aligned}
\sigma(\mathcal{R})_{\text{All}} &= 0.0051 \text{ Pa.min/L}, & \sigma(\mathcal{R})_3 &= 0.0041 \text{ Pa.min/L}, \\
\sigma(V_o)_{\text{All}} &= 0.0085 \text{ m/s}, & \sigma(V_o)_3 &= 0.0026 \text{ m/s}, \\
\sigma(\eta)_{\text{All}} &= 7.94\%, & \sigma(\eta)_3 &= 0.44\%,
\end{aligned}
\tag{1}
$$

where $\sigma(\cdot)_{\text{All}}$ denotes the standard deviation across all six grids and $\sigma(\cdot)_3$ denotes the standard deviation of the simulated data from the last 3 grids (i.e., cases with 2.0, 2.5, and 3.0 million unstructured tetrahedral cells). Based on the asymptotic convergence trend observed in these final three cases, the intermediate 2.5-million cell grid resolution was selected for the overall study. The choice is consistent with detailed studies on grid convergence and computational stability for simulating physiologically realistic airflow and particle deposition in the human respiratory system; e.g., see [46].

Notably, while using the LES scheme, a finer mesh refinement typically leads to more precise outcomes. For instance, at the upper limit, achieving results akin to Direct Numerical Simulation (DNS) is possible if the grid dimensions are smaller than or comparable to the Kolmogorov length scale, $\mathcal{K}$, representing the smallest, most dissipative eddies in a turbulent flow and defined as [47]:

$$
\mathcal{K} = \left(\frac{\nu^3}{\varepsilon}\right)^{1/4},
\tag{2}
$$

with $\varepsilon$ as the turbulence dissipation rate and $\nu$ being the fluid kinematic viscosity. Another length scale of note is the Taylor scale, $\lambda$, which typically exceeds $\mathcal{K}$ and is defined as:

$$
\lambda = \left(\frac{10\nu k}{\varepsilon}\right)^{1/2},
\tag{3}
$$

with $k$ being the turbulence kinetic energy. From the simulation data of the inhaled airflow field, it is seen that both $\lambda$ and $\mathcal{K}$ collapse to $\mathcal{O}(10^{-4})$ m, while the mean grid scale is also $\rightarrow \mathcal{O}(10^{-4})$ m, suggesting that the test grid has been sufficiently resolved for reliable estimation of the transport parameters.

**Numerical simulation of inhaled airflow.** We have implemented the LES approach to numerically model the inhaled airflow, with eddies exceeding the grid scale explicitly resolved, whereas those that are smaller than the grid scale are approximated. Specifically, fluctuations below the grid size, referred to as subgrid scales, are filtered out, and their impact on larger scales is replicated through modeling. Assuming incompressible, isothermal conditions for the inhaled air flux,

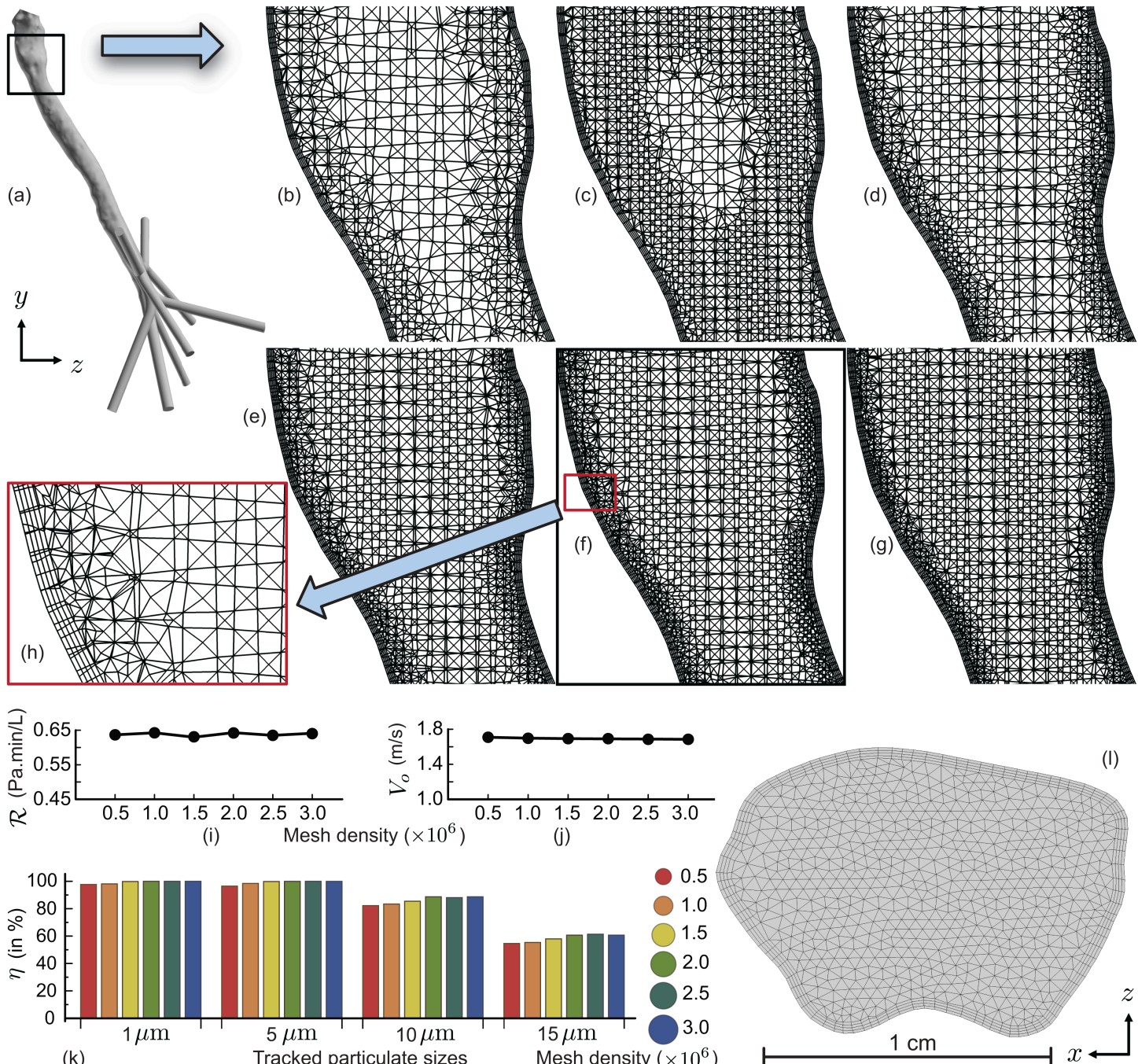

**Fig 2. Assessing the numerical sensitivity.** Panel (a) marks the location of the demonstrated unstructured tetrahedral meshes spatially refined with the following cell counts (in million): (b) 0.5, (c) 1.0, (d) 1.5, (e) 2.0, (f) 2.5, (g) 3.0. Panel (h) highlights the four layers of pentahedral cells used for near-wall refinement. Panels (i)-(k) present the computed flow and particulate transport variables across the six mesh resolutions: (i) resistance $\mathcal{R}$ to inhaled airflow, (j) area-weighted average airflow velocity $V_o$ at the outlet faces, and (k) deposition (or, penetration) rate $\eta$ (%) in bronchial pathways for select particle sizes of 1, 5, 10, and 20 $\mu$m. The circle sizes to the right of panel (k) represent the number of tracked particulates in each simulation, scaled proportionally: red = 597, orange = 831, ocher = 1208, green = 1459, dark green = 1622, and blue = 1954 tracked particulates per size. In each case, the tracked particle count is equal the number of mesh facets (faces) on the inlet surface, directly proportional to the spatial resolution of the respective mesh. Panel (l), with its length scale included at the bottom, shows a magnified top view of the inlet surface for the mesh in (f), consisting of 400 quadrilateral facets and 1222 triangular facets.

the filtered continuity and Navier–Stokes equations are respectively:

$$\frac{\partial \overline{u}_i}{\partial x_i} = 0 \tag{4}$$

and

$$\frac{\partial \overline{u}_i}{\partial t} + \frac{\partial}{\partial x_j}\left(\overline{u}_i \overline{u}_j\right) = -\frac{1}{\rho}\frac{\partial \overline{p}}{\partial x_i} + \frac{\partial}{\partial x_j}\left(\nu \frac{\partial \overline{u}_i}{\partial x_j}\right) - \frac{\partial \tau_{ij}}{\partial x_j}. \tag{5}$$

Eq 5 describes the conservation of momentum in each spatial direction ($i \in 1, 2, 3$). Here $\overline{u}_i$ represents the filtered (i.e., resolved) velocity, $\overline{p}$ is the filtered pressure, $\nu$ and $\rho$ are respectively the kinematic viscosity and the density of inhaled warmed-up air, and $\tau_{ij}$ is the subgrid scale (SGS) stress tensor defined by

$$\tau_{ij} - \frac{1}{3}\tau_{kk}\delta_{ij} = -\nu_{sgs}\left(\frac{\partial \overline{u}_i}{\partial x_j} + \frac{\partial \overline{u}_j}{\partial x_i}\right), \tag{6}$$

where $\nu_{sgs}$ is the SGS kinematic viscosity and $\delta_{ij}$ is the Kronecker delta. It is to be noted that $\tau_{kk}$, which comprises the isotropic part of the SGS stresses, is not modeled but added to the filtered static pressure. Subsequently, the instantaneous field velocity is given by

$$u_i = \overline{u}_i + u_i^{sgs}, \tag{7}$$

with $u_i^{sgs}$ representing the SGS velocity fluctuations. The flow patterns and particle dispersion within the human respiratory system will be strongly impacted by the secondary flows common in such complex geometries and by the airflow transitions between laminar and turbulent regimes. To simultaneously capture the transitional features as well as the secondary flow formations, this study uses the dynamic subgrid-scale kinetic energy transport model [48–50]. Therein the SGS kinematic viscosity, $\nu_{sgs}$, is obtained from the Kolmogorov-Prandtl hypothesis [51] in the following form:

$$\nu_{sgs} = C_k \, k_{sgs}^{1/2} \, \Delta_f. \tag{8}$$

In the above equation, $C_k$ is a constant value and $\Delta_f$ is the filter size computed as $\Delta_f \equiv$ (grid cell volume)$^{1/3}$. The term $k_{sgs}$ represents the SGS kinetic energy, defined by

$$k_{sgs} = \frac{1}{2}\left(\overline{u_i u_j} - \overline{u}_i \overline{u}_j\right). \tag{9}$$

To derive $k_{sgs}$, we solve the following filtered transport equation:

$$\frac{\partial k_{sgs}}{\partial t} + \frac{\partial}{\partial x_j}\left(k_{sgs}\overline{u}_j\right) = \frac{\partial}{\partial x_j}\left(\nu_{sgs}\frac{\partial k_{sgs}}{\partial x_j}\right) + \frac{\partial \overline{u}_i}{\partial x_j}\left[\nu_{sgs}\left(\frac{\partial \overline{u}_i}{\partial x_j} + \frac{\partial \overline{u}_j}{\partial x_i}\right) - \frac{2}{3}k_{sgs}\delta_{ij}\right] - C_\varepsilon \frac{k_{sgs}^{3/2}}{\Delta_f}, \tag{10}$$

with the model constants in the previous equations, i.e., $C_k$ and $C_\varepsilon$, being determined dynamically [48].

Applying the LES scheme as described above, the inhaled airflow through the anatomical airspace was replicated on ANSYS Fluent 2024 R1 for 15 L/min inhalation rate, conforming with normal relaxed breathing conditions [52]. See Fig 1c for the pressure inlet face at the location of vocal folds in the larynx and for the pressure outlets at the distal ends of the reconstructed generation 3 bronchial tubes. Enforcing no-slip (i.e., zero velocity) boundary condition at the airway walls, the pressure gradient-driven simulation was launched with gauge pressures of 0 Pa at the inlet and -50 Pa at the outlets

with a targeted airflow rate of 15 L/min passing through the cavity. Note that the solver did not impose a parabolic velocity profile at the inlet; rather it solved the full Navier-Stokes equations with the applied boundary conditions, and the resulting velocity profile was a result of the simulation, not an initial assumption. Eventually, post-processing of the simulated flow field revealed a total pressure gradient of 10.16 Pa driving the flow through the redacted geometry. The numerical scheme used time-steps of 0.0002 s, for a flow solution time of 0.35 s. The latter was preliminarily chosen based on reported findings [50] on the time-step size warranted to fully resolve the unsteady turbulent airflow field in a realistic upper airway model, it being sufficiently small and order-wise comparable to the Kolmogorov time scale $= (\nu/\varepsilon)^{1/2}$ [47]. To confirm the feasibility of the applied time-step size, additionally assessed was the Courant number, also known as the *CFL* (Courant–Friedrichs–Lewy) number [53], which is a key stability criterion in computational fluid dynamics and is evaluated here as:

$$CFL = \frac{V_o\,\Delta\tilde{t}}{s}. \tag{11}$$

In the above, $V_o$, being the average air velocity through the bronchial outlets (marked in Fig 1c-d), is estimated as $Q/A_o$, where $Q$ is the target volumetric flow rate of 15 L/min and $A_o = 148.65 \times 10^{-6}$ m$^2$ is the total open cross-sectional area at the outlets (the latter is extracted from the digitized geometry). Next, $s = 4.73 \times 10^{-4}$ m is the mean spatial grid size in the tetrahedral mesh modeling the bulk cavity. It is calculated as $(6\sqrt{2}\,\Omega)^{1/3}$ (using the volume formulation for a tetrahedron), where $\Omega$ represents the mean cell volume calculated by dividing the total airspace volume by the net number of tetrahedral elements in the built mesh. Finally, $\Delta\tilde{t}$ is the implemented time-step size. Using these parameters in consistent units, the estimated *CFL* is approximately 0.71. The condition *CFL*<1 indicates that the flow travels less than one grid cell per time-step, which is favorable to numerical stability, particularly in explicit integration schemes, and suggests that the chosen time-step is appropriate for reliably capturing the complex flow dynamics without introducing numerical inconsistencies.

The simulation was executed on a segregated solver with pressure-velocity coupling and second-order upwind spatial discretization. For the transient formulation, a bounded second-order implicit scheme was employed, to strike a balance between accuracy (due to the second-order formulation), stability (from the implicit approach), and boundedness (to prevent non-physical oscillations). The eventual solution convergence was monitored by minimizing the mass continuity residual to $\mathcal{O}(10^{-3})$ and the velocity component residuals to $\mathcal{O}(10^{-6})$. Also, considering the warmed-up state of inhaled air passing through the respiratory pathway, the air density $\rho$ was set at 1.204 kg/m$^3$ in the simulations, with $15.16 \times 10^{-6}$ m$^2$/s as its kinematic viscosity $\nu$.

**Assessment of mesh quality in relation to simulated flow conditions.** Physically, wall y+ represents a dimensionless measure of the distance from the first cell center to the wall, normalized by the viscous length scale [54]. It indicates whether that location lies within the viscous sublayer, buffer layer, or logarithmic layer of the boundary layer. This parameter is useful for assessing the mesh quality near walls, particularly within the boundary layer where viscous effects dominate. In our test domain, the flow simulations yielded the following results: (a) a mean wall y+ of 0.796 when considering the entire airspace boundary (excluding the inlet and outlets, see Fig 1); and (b) a mean wall y+ of 0.825 when focusing only on the bronchial region. These values suggest that the mesh near the wall is well within the typical range for boundary layer resolution, as the target is usually y+ < 1 for accurate near-wall modeling. This adequate boundary layer resolution supports the reliability of the forthcoming numerical results.

In addition, the ICEM mesh quality (for the grid selected from the sensitivity analysis) was evaluated. Only 0.283% of the total elements fell into the lowest quality bin, which ranges from 0.35 to 0.38. The low percentage in this bin, combined with higher quality scores for the majority of the mesh elements and the stable convergence trends in the flow simulation, indicates that the overall mesh quality is acceptable.

**Numerical experiments for particulate transport.** Within the solved inhaled airflow field, the downwind motion of particulates formed intra-URT was tracked from the vocal fold region of the larynx (see the marked inlet face in Fig 1c). The air-particle phases were one-way coupled with the particles (assumed spherical) being impacted by the ambient

flow field; the underlying flow domain was considered quasi-steady while evaluating the particle transport parameters. Lagrangian-based inert discrete phase model, with a Runge-Kutta solver, was used to numerically integrate the particle transport equation:

$$\frac{du_{pi}}{dt} = \frac{18\mu}{d^2\rho_p} \frac{C_D Re_p}{24} \left(u_i - u_{pi}\right) + g_i \left(1 - \frac{\rho}{\rho_p}\right) + F_i. \tag{12}$$

Here $u_{pi}$ represents the particulate velocity, $\rho_p$ is the material density of the particulates, $d$ represents the particulate diameter, $Re_p$ is the particulate Reynolds number, $g_i$ signifies the gravitational acceleration in the $i$ direction, $C_D$ is the drag coefficient, and $F_i$ represents additional body forces per unit particulate mass in the form of the Saffman lift force exerted by a typical flow-shear field on small particulates transverse to the airflow direction. In this context, note that the solution scheme considered the particulates to be large enough to ignore any Brownian motion effects on their dynamics.

To evaluate the drag component in Eq 12, the quantities $Re_p$ and $C_D$ are respectively computed as: $Re_p = \rho_p d |u_i - u_{pi}| /\mu$ and $C_D = a_1 + a_2/Re_p + a_3/Re_p{}^2$; where $\mu$ is the molecular viscosity of the ambient fluid (i.e., air), while $a_1$, $a_2$, and $a_3$ are functions of $Re_p$ determined based on the spherical drag law [55]. Subsequently, the particulate trajectories are derived from their spatiotemporal locations, $x_i(t)$, obtained by numerical integration of the following velocity vector equation:

$$u_{pi} = \frac{dx_i}{dt}. \tag{13}$$

Considering stationary walls with no-slip and through implementation of 'trap' discrete phase model boundary condition, intra-airway aerial tracking of a particulate pathline was stopped once it entered the mesh element layer adjacent to the enclosing walls of the respiratory cavity. To note in addition, the discrete phase boundary condition types were 'reflect' for the inlet in the redacted geometry and 'escape' at the outlets. The particulates escaping through the bronchiolar outlets were also recorded, and represented the particulates penetrating to the respiratory bronchioles and deep lungs. The numerical experiments in this study tested particulates of diameters $1 - 30\,\mu m$ (with increments of $1\,\mu m$). In total, $\mathcal{N}$ = 1622 particulates of each size were tracked; $\mathcal{N}$ being the number of mesh facets (faces) that made up the inlet surface (see Fig 2l), with the starting locations of the particulates being at the centroids of the facets. Therefore, the starting distribution of the monodisperse particulates was identical to the non-uniform mesh facet distribution at the laryngeal inlet plane, with a higher density near the edges (see Fig 2l). This fortuitously aligned with the hypothesis that these are the particulates formed from mucosal fragmentation at the URT surface and are expected to crowd the near-wall space. Subsequently, the eventual particulate distribution and the pressure-driven airflow velocity profile at the geometry outlets emerged naturally as solutions to the governing fluid dynamics equations under the imposed boundary conditions. Also, considering that saliva-mixed mucus is 99.5% water [56], the material density of the particulates was assumed to be $\rho_p$ = 1.00175 g/mL, approximated as the weighted average (see Eq 14 below) between 99.5% water and the residual pathogenic and biological non-volatile compounds with a representative density of $\rho_{nv}$ = 1.35 g/mL; e.g., for protein [57], the density being independent of the nature of the protein and particularly independent of its molecular weight. The weighted average calculation for the simulated particulate density is as follows:

$$\rho_p = \frac{0.5 \times \rho_{nv} + 99.5 \times \rho_w}{100}, \tag{14}$$

with $\rho_w$ = 1.0 g/mL representing the density of water. The consideration of inertness for the tracked particulates (per Eq 12) also implies that the modeling framework was agnostic to the biological nuances of the embedded constituents in the particulates, beyond the imposition of the appropriate physical properties, e.g., density. The modeling approach also

discounted any heat transfer effects between the flow constituents and the surrounding tissues enclosing the anatomical airspace.

**Supplementary 'diversion': Bronchial transmission trend for particulates inhaled from outside.** While the focus of this study is singularly on the downwind bronchial transmission of particulates generated within the URT, it would be of scholarly interest to undertake a brief detour at this point to explore an auxiliary question (that has been nonetheless well-explored in literature): for the particulates inhaled from the external environment, which particulate sizes would be efficient at penetrating to the bronchial spaces? Here, we address this question specifically for the test geometry used in this study, through transport simulations within its non-redacted complete anatomy (Fig 1b) wherein inhaled particulates are entering the airway through both the left and right nostrils.

Comparing the cavity spaces in the complete airway geometry (Fig 1b) and the redacted model (Fig 1c-d), the volume of the former is approximately 3.48 times larger than that of the redacted system. Accordingly, following the described meshing protocol, the full airway was discretized into 2.5 million $\times$ 3.48 = 8.7 million unstructured tetrahedral elements, with four layers of pentahedral cells lining the cavity surfaces. Using this meshed geometry, particulate transport was simulated at an inhaled airflow rate of 15 L/min, following methods outlined earlier. The only modification was setting the material density of the inhaled particulates to 1.3 g/mL, consistent with typical values for environmentally dehydrated respiratory ejecta [26,56], which constitutionally form the pathogen-bearing particulates inhaled from external air.

Fig 3 demonstrates the bronchial transmission trend for particulates inhaled from outside. Therein, panel (a) presents $\eta_d$, the penetration rate through the bronchiolar outlets, indicating the proportion of inhaled particulates reaching the deeper lung regions. Panel (b) shows the cumulative transmission percentage, $\eta$, summing up the deposition rates at the primary, secondary, and tertiary bronchi, together with the penetration rate (in %) through the geometry outlets that serve as entry to the respiratory bronchioles. Considering select particulate sizes, $\eta$ starts at 99.36% for particulates of diameter $d = 1$ $\mu$m, decreases to 96.86% for $d = 5$ $\mu$m, to 31.46% for $d = 10$ $\mu$m, and approaches 0% for $d = 15$ $\mu$m and larger. Remarkably though, as we will find later (in the Results), for particulates generated *within* the URT, the quantity $\eta$ shoots up significantly, approaching 90% for $d = 10$ $\mu$m and steadily sustaining at > 60% even for $d = 15$ $\mu$m.

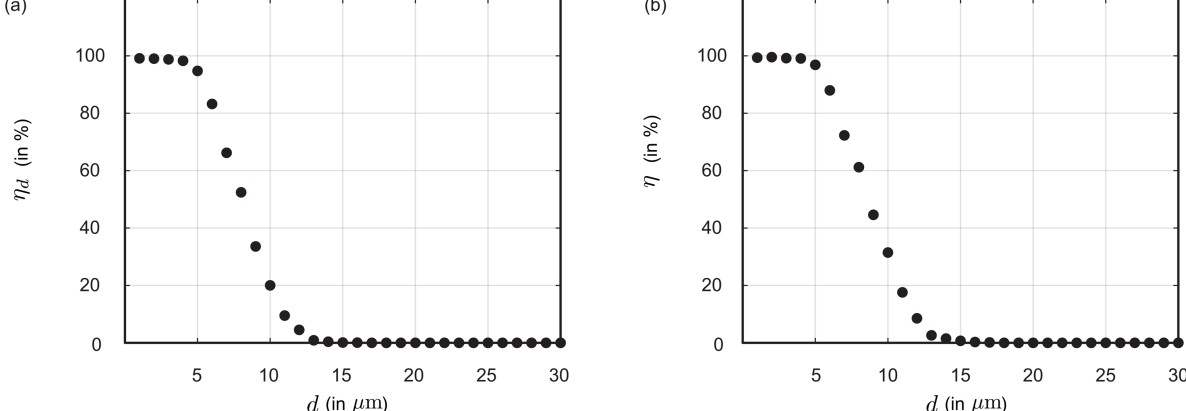

**Fig 3**. **Transmission trend for particulates inhaled from outside and navigating the complete anterior airway.** (a) $\eta_d$ isolates the penetration rate through the bronchiolar outlets with the tracked particulates moving into the deeper lung regions; (b) $\eta$ represents the cumulative deposition percentages at the primary, secondary, and tertiary bronchi, together with the penetration rate through the bronchiolar outlets into the deeper lung region. The reader may find it insightful to compare the trend reported here with Fig 5a,e on the navigation trend for particulates generated within the URT (and still airborne at the larynx). In the latter scenario, the larger particulates (e.g., ones $\gtrsim 10$ $\mu$m) exhibit much greater efficiency at penetrating to the lower airway.

## A simplified analytical model for inhaled transport through the laryngotracheal space

To verify the particulate transport trends to the bronchial domain as derived numerically, let us invoke a simulation-informed reduced-order analytical model (S-ROAM), with a two-dimensional channel mimicking the laryngotracheal domain. Therein, the inhaled airflow is modeled in the dimensionless complex $\chi$ plane, with $\chi = \alpha + i\beta$ and $i\beta$ aligned with the channel's streamwise axis; see Fig 4. The two-dimensional channel has its streamwise length and cross-stream width based on the averaged dimensions of the anatomical tract from Fig 4a-b. The channel walls are placed at $\alpha = 0, 1$. Based on the geometric inputs, the S-ROAM enforces $L/W \approx 9$, where $L$ represents the streamwise channel length and $W$ is the inlet width. For a detailed exposition of this classical modeling approach for complex respiratory systems, see our recent preprint [58].

The reduced system modeled the core of an extended vortex patch emerging in the full-scale numerical simulation as a random collection of five point vortices distributed over the area of the simulated vortex core (on the channel plane, mapping the mid-sagittal section of the three-dimensional cavity; see Fig 4b), bearing dimensionless circulations $\Gamma_i = \omega A/5UW$, where $\omega$ was the mean vorticity (as determined from the numerical simulations) spread over the patch area $A$ and $U$ is the characteristic streamwise air speed through the channel. From the simulated data, we had: $\omega \approx 1750\ \text{s}^{-1}$, $A \approx 2.65 \times 10^{-5}\ \text{m}^2$, and $U = Q/A_m = 2.16$ m/s, with $A_m = 115.82\ \text{mm}^2$ being the area of the channel inlet

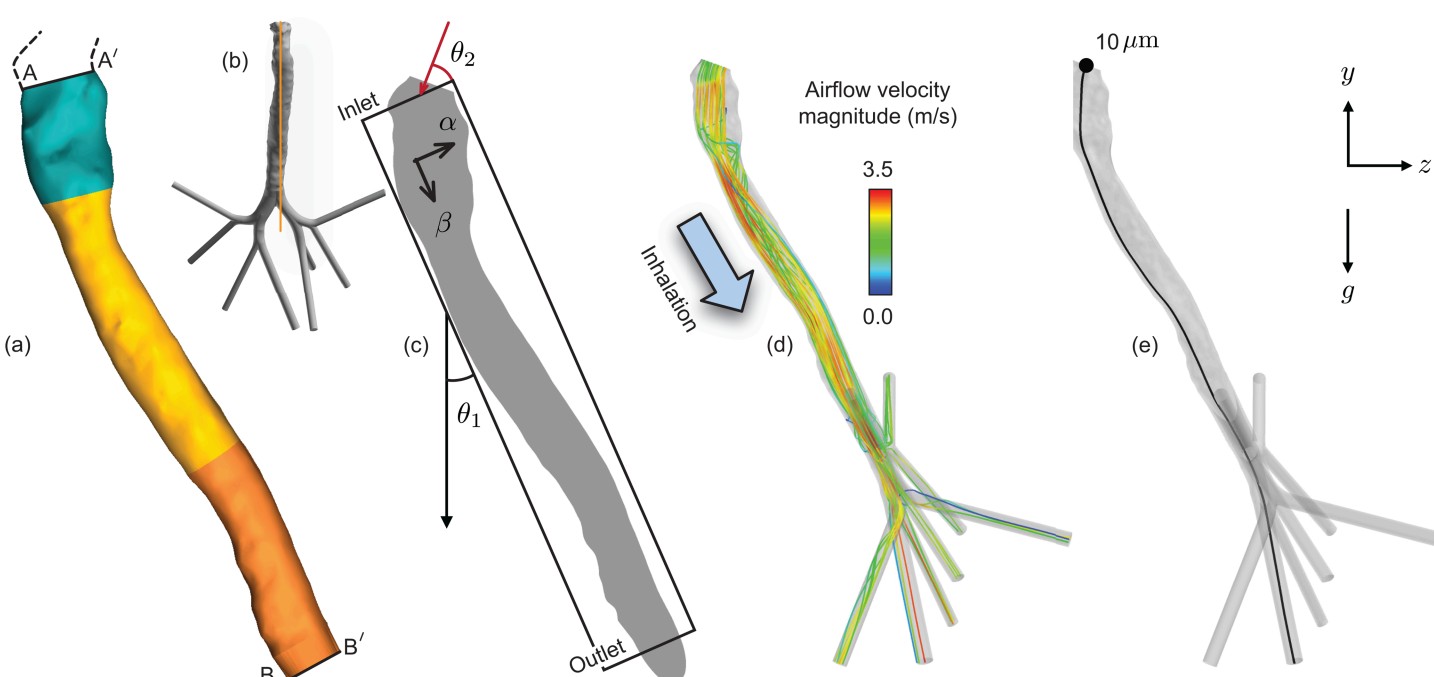

**Fig 4. Defining the analytical domain, with representative numerical visuals.** (a) Laryngotracheal region upwind from the primary bronchi used to develop a simulation-informed reduced-order analytical model (S-ROAM) for bronchial transmission. AA′ marks the larynx upwind face and BB′ marks the downwind face of the distal tracheal cavity; see labels in Fig 1d. The cross-sectional areas of the cavity at AA′ and BB′ are 108.61 mm² and 123.03 mm², respectively. The corresponding hydraulic diameters are 11.03 mm and 12.20 mm, respectively. The linear spatial distance between the two faces is approximately 115.84 mm, at angle $\theta_1 = 23.77°$ to the vertical (direction of gravity in the numerical simulations). Panel (b) highlights the location and orientation of the planar cross-section shown in panel (c) and also previously as the location of the mesh visuals in Fig 2. The vortex strengths and positions for the S-ROAM are extracted from the simulated data mapping this two-dimensional plane; the plane cuts through the entire cavity of panel (a). $\theta_2$ = 33.94° is the assumed angle in S-ROAM, at which microdroplets enter the AA′ inlet face and is governed by the anatomical shape of the cavity upwind from AA′ (marked by the dashed black traces; also see Fig 1). (d) 30 randomly selected representative velocity streamlines extracted from the numerically simulated inhaled airflow field. (e) Simulated trajectory of a representative 10-$\mu$m particulate. The geometric centroid of the inlet face, marked by the solid circle, was chosen as the particulate's position at tracking time $t = 0$.

face (averaged between the CT-derived cross-sectional areas at AA′ and BB′ in Fig 4a) and $Q = 15$ L/min $= 0.00025$ m$^3$/s being the simulated inhalation flux. Based on the angular orientation of the modeled core in the simulated vorticity field, the model point vortices were placed on a line that subtended $\approx 2.2°$ in counter-clockwise sense with the vertically downward direction. The anatomical planar section from which the vorticity information was extracted is shown in Fig 4c.

In the S-ROAM, the straight streamlines at the channel walls were established through inclusion of periodic images of the point vortices in the cross-stream $\alpha$ direction. The resulting (dimensionless) complex potential for this system, with a dimensionless background speed of unity, is [59–61]:

$$F(\chi) = \phi(\alpha, \beta) + \mathrm{i}\,\psi(\alpha, \beta) = \mathrm{i}\,\chi + \sum_{i=1}^{5} \frac{\Gamma_i}{2\pi\mathrm{i}} \log\left\{ \frac{\sin\left[\pi(\chi - \chi_i)/2\right]}{\sin\left[\pi(\chi + \chi_i^*)/2\right]} \right\}, \tag{15}$$

with $\phi$ as the velocity potential and $\psi$ as the real-valued flow streamfunction; the asterisk denotes complex conjugation. Subsequently, the inhaled particulate motion was analytically derived using a simplified version of the Maxey-Riley equation [62,63], in the two-dimensional vector form [64,65]:

$$\frac{d\boldsymbol{w}}{dt} = -\left[ \boldsymbol{J} + \frac{2\,Stk^{-1}}{3(\sigma + 1)}\boldsymbol{I} \right] \cdot \boldsymbol{w} + \frac{\sigma}{\sigma + 1}\left( Fr^{-2}\boldsymbol{g} - \frac{\mathrm{D}\boldsymbol{u}}{\mathrm{D}t} \right). \tag{16}$$

Here $\boldsymbol{u}$ is the local fluid velocity in vector coordinates, $\boldsymbol{w} = \boldsymbol{u_p} - \boldsymbol{u}$ is the relative velocity of a particulate with velocity $\boldsymbol{u_p}$, $\boldsymbol{J}$ is the two-dimensional Jacobian matrix, $\boldsymbol{I}$ is the identity matrix, $\boldsymbol{g}$ implies gravity, with the nondimensional parameters being Stokes number $Stk \equiv d^2 U/18\nu L$, Froude number $Fr \equiv U/\sqrt{gL}$, and $\sigma \equiv 2\left(\rho_p - \rho\right)/3\rho$.

Although viscous effects are neglected in the above velocity model, particle drag is included. This form of the model ignores the Faxen correction terms and the Basset-Boussinesq history force. Particulates are assumed to be entering the S-ROAM domain (inclined at an angle $\theta_1 = 23.77°$ to the direction of gravity, per the general shape and orientation of the test anatomical cavity) with speed $U$ and at an angle of $\theta_2 = 33.94°$ with respect to the S-ROAM's $\alpha$-axis, mimicking the simulated motion of particles entering the region; see Fig 4c. As representative examples, particulate trajectories were derived for two initial positions at the mid-point and near the right edge of the reduced channel's inlet span, and for seven different particulate diameters, namely 5, 10, 15, 20, 25, 30, and (as an extremal case) 50 $\mu$m.

**Translational integration: Connecting the fluid dynamics outcomes to pathogen-specific virological data**

From the numerical experiments, this study has deduced the simulated deposition and penetration efficiencies of the URT-derived particulates along the bronchial tubes (see Table 1) and evaluated the respective volumetric transmission to the bronchi. The projected net deposited (at the primary, secondary, and tertiary bronchi) and penetrated (moving into the respiratory bronchioles) volumes are then multiplied with the sputum viral concentration for a specific pathogen (in this study, SARS-CoV-2) to evaluate and compare the viral load transported via inhaled aerial advection of URT particulates to the lower airway and deep lungs, for select test particulate sizes, namely, 1, 5, 10, and 15 $\mu$m in Table 2, and more comprehensively later on, in Fig 5.

Note that the average sputum viral concentration for SARS-CoV-2 has been reported as $\mathcal{V} = 7.0 \times 10^6$ virions/ml, through count measurements of the RNA copies for the single-stranded virus present in the airway liquid samples collected from hospitalized COVID-19 patients; see [25].

## Nomenclature for selected mathematical symbols

| Symbols | Description |
| --- | --- |
| $Q$ | Volumetric inhaled airflow rate |
| $\eta$ | Cumulative bronchial transmission %, for each particulate size |
| $\eta_p$ | Deposition % in the primary bronchi, for each particulate size |
| $\eta_s$ | Deposition % in the secondary bronchi, for each particulate size |
| $\eta_t$ | Deposition % in the tertiary bronchi, for each particulate size |
| $\eta_d$ | Penetration % beyond tertiary bronchi, for each particulate size |
| $V_o$ | Air velocity magnitude at the geometry outlets |
| $A_o$ | Net outlet opening area |
| $A_{in}$ | Area of inlet face in redacted geometry |
| $\sigma$ | Standard deviation operator |
| $\mathcal{K}$ | Kolmogorov length scale |
| $\lambda$ | Taylor length scale |
| $\nu$ | Fluid kinematic viscosity |
| $\varepsilon$ | Turbulence dissipation rate |
| $k$ | Turbulence kinetic energy |
| $\rho$ | Fluid density |
| $u_i$ | Instantaneous flow velocity component |
| $\bar{u}_i$ | Filtered (resolved) velocity |
| $\bar{p}$ | Filtered pressure |
| $\tau_{ij}$ | Subgrid scale stress tensor |
| $\delta_{ij}$ | Kronecker delta |
| $\Delta\tilde{t}$ | Simulation time-step size |
| $u_{pi}$ | Inhaled particulate velocity |
| $d$ | Particulate diameter |
| $\rho_p$ | Material density in particulates |
| $\rho_{nv}$ | Density of non-volatile compounds in saliva-mixed mucus |
| $\rho_w$ | Water density |
| $\chi$ | Dimensionless complex plane in analytical setup |
| $\Gamma_i$ | Dimensionless circulation |
| $\omega$ | Mean vorticity in the simulated field |
| $\mathcal{N}$ | Total number of particulates numerically tracked for each diameter |
| $T$ | Number of days over which viral load transmission is estimated |
| $\mathcal{P}$ | Particle density size function operator |
| $\delta t$ | Duration of one breathing cycle |
| $\mathcal{V}$ | Sputum virion concentration |
| $V_L$ | Viral load transmitted to bronchial region |
| $I_D$ | Infection-triggering viral load (Infectious dose) |
| $\tau$ | Tortuosity |

**Table 1. Numerical transmission trend: The numerically simulated bronchial deposition and penetration data are detailed herein. Particulate sizes explored further in Table 2 (to formalize the viral transmission trends as a function of the microdroplet dimensions) are in bold font.** *Symbols: d =* tested aerosol (or, droplet) diameter; $\mathcal{N}$ = total number of aerosols (or, droplets) tracked for each particulate diameter; $n_p$ = number of deposited particulates in the primary bronchi; $n_s$ = number of deposited particulates in the secondary bronchi; $n_t$ = number of deposited particulates in the tertiary bronchi; $n_d$ = number of particulates penetrating into the respiratory bronchioles toward the deep lungs; $\eta$ = cumulative deposition (or, penetration) rate (in %) to the bronchial pathways and computed as $100 \times \left( \sum_j n_j \right) / \mathcal{N}$, with $j \in \{p, s, t, d\}$.

| Separated particulate size ($d$, in $\mu$m) | $\mathcal{N}$ | $n_p$ | $n_s$ | $n_t$ | $n_d$ | Bronchial transmission ($\eta$, in %) |
|---|---|---|---|---|---|---|
| **1** | **1622** | **5** | **1** | **0** | **1616** | **100.00** |
| 2 | 1622 | 16 | 17 | 3 | 1586 | 100.00 |
| 3 | 1622 | 9 | 2 | 0 | 1611 | 100.00 |
| 4 | 1622 | 7 | 7 | 1 | 1607 | 100.00 |
| **5** | **1622** | **8** | **25** | **3** | **1586** | **100.00** |
| 6 | 1622 | 18 | 35 | 10 | 1559 | 100.00 |
| 7 | 1622 | 26 | 65 | 34 | 1493 | 99.75 |
| 8 | 1622 | 38 | 98 | 34 | 1422 | 98.15 |
| 9 | 1622 | 57 | 111 | 51 | 1277 | 92.23 |
| **10** | **1622** | **80** | **134** | **39** | **1175** | **88.04** |
| 11 | 1622 | 85 | 144 | 55 | 1083 | 84.28 |
| 12 | 1622 | 119 | 181 | 56 | 930 | 79.28 |
| 13 | 1622 | 160 | 208 | 59 | 787 | 74.85 |
| 14 | 1622 | 216 | 208 | 60 | 634 | 68.93 |
| **15** | **1622** | **225** | **224** | **93** | **453** | **61.34** |
| 16 | 1622 | 225 | 220 | 90 | 321 | 52.77 |
| 17 | 1622 | 217 | 216 | 111 | 217 | 46.92 |
| 18 | 1622 | 192 | 203 | 108 | 146 | 40.01 |
| 19 | 1622 | 192 | 169 | 104 | 72 | 33.11 |
| 20 | 1622 | 142 | 167 | 86 | 45 | 27.13 |
| 21 | 1622 | 95 | 173 | 63 | 25 | 21.95 |
| 22 | 1622 | 59 | 159 | 49 | 13 | 17.26 |
| 23 | 1622 | 39 | 129 | 22 | 10 | 12.33 |
| 24 | 1622 | 15 | 103 | 4 | 1 | 7.58 |
| 25 | 1622 | 12 | 71 | 0 | 0 | 5.12 |
| 26 | 1622 | 10 | 48 | 0 | 0 | 3.58 |
| 27 | 1622 | 8 | 30 | 0 | 0 | 2.34 |
| 28 | 1622 | 4 | 15 | 0 | 0 | 1.17 |
| 29 | 1622 | 2 | 1 | 0 | 0 | 0.18 |
| 30 | 1622 | 0 | 0 | 0 | 0 | 0.00 |

**Table 2. Magnitude of viral load transmission into bronchial pathways as a function of inhaled particulate sizes: The duration in consideration is 3 days, with a conservative estimate of 1 particulate of each test size generated during each breathing cycle. The breathing cycles last 5 s [68]. Listed data is a subset of Fig 5f.**

| Separated particulate size ($d$, in $\mu$m) | Number of virions ferried to the bronchial pathways ($L_v$) |
|---|---|
| 15 | 394 |
| 10 | 167 |
| 5 | 24 |
| 1 | 0 |

## Results

### Bronchial deposition and lower airway penetration as projected from numerical modeling

Fig 4d presents sample inhaled airflow velocity streamlines from the numerical simulation of inhaled airflow. The simulated flux of 15 L/min (to be exact, 14.97 L/min in the reported simulation) warranted an inlet-to-outlet static pressure gradient of −9.51 Pa, with the total pressure gradient driving the flow being 10.16 Pa. Both the measurements were area-weighted averages across the cross-sections. Within the inhaled airflow field, the test particulates were tracked with their

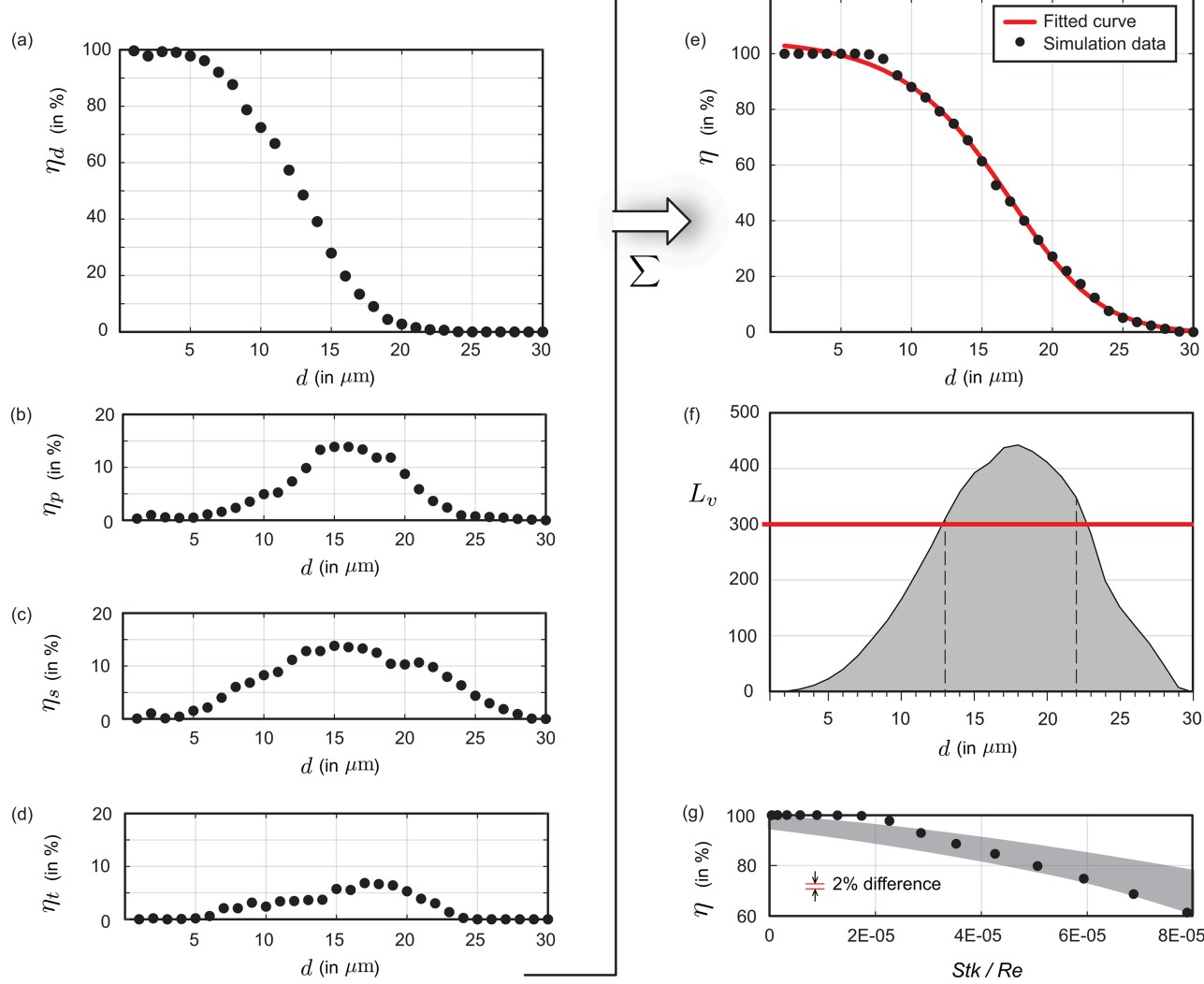

**Fig 5. Numerically simulated bronchial transmission trend for URT-derived particulates navigating through the redacted test geometry.** (a) $\eta_d$ represents the penetration rate through the bronchiolar outlets, calculated as $100 \times n_d/\mathcal{N}$ %, as a function of the tracked particulate sizes $d$. (b)-(d) Respective deposition percentages $\eta_p$, $\eta_s$, and $\eta_t$ at the primary, secondary, and tertiary bronchi. They are computed as $100 \times n_j/\mathcal{N}$, with $j \in \{p, s, t\}$; see Table 1 caption for definitions of $n_j$. (e) $\eta$ represents the cumulative deposition percentages at the primary, secondary, and tertiary bronchi, summed with the penetration rate through the bronchiolar outlets into the deeper lung regions. The underlying solid red line is a fitted curve of the Heaviside step function form (see Eq 18), with the solid black circles showing simulation-derived $\eta$ values (reported in Table 1). (f) $L_v$ provides a count of virions penetrating to the bronchial space in 3 days, with a conservative estimate of 1 particulate of each test size generated during each breathing cycle. For perspective, the red line marks the infection-triggering viral load for SARS-CoV-2 [26,27]. Selected values from this plot are listed in Table 2. (g) Comparison of the numerical penetration trend with published experimental data [66]. The shaded patch marks the experimental observation domain, while the data points from the present study are represented by solid black circles.

starting locations at the centroids of mesh facets on the inlet surface of the redacted geometry. The initial positions of the particulates hence coincide with the cross-sectional space spanning the vocal fold region of the laryngeal cavity (commensurate with their hypothesized formation sites at the URT—through breakup of mucus strata along the nasopharynx, the oropharynx, and the vocal folds—and their subsequent aerial locations). A representative particulate trajectory for $d = 10$ $\mu$m has been additionally shown in Fig 4e. At tracking time $t = 0$, the particulate was assumed to be positioned

at the geometric centroid of the inlet face. Eventually, the sample particulate penetrates through the outlet to respiratory bronchioles, thereby moving into the deeper lung recesses.

Fig 5a-e and Table 1 detail the lower airway deposition and penetration data for the tested particulates bearing diameters $1 - 30\,\mu$m (with increments of $1\,\mu$m). While it is expected that the particulates $\lesssim 5\,\mu$m would comfortably penetrate to the deeper regions of lungs (as is clearly the case per Table 1; see the top rows), the high transmission percentages, e.g., for even the 10- and 15-$\mu$m particulates is striking—them being 88.04% and 61.34%, respectively.

## Viral load transmitted to the lower airway

Table 2 lists the viral load transmitted to the bronchial pathways, for the representative test particulate sizes 1, 5, 10, and 15 $\mu$m. A more comprehensive depiction of the transmitted viral loads for the test particulate sizes is in Fig 5f. For the time scale $T$ (in days) over which the viral load transmission is to be estimated, we have used 3 days based on reported data [67] on the typical time interval that has revealed confirmed infection onset in the deep lungs subsequent to the emergence of initial symptoms along the URT. Further, applying $\delta t$ = 5 s as the average duration for a complete breathing cycle [68], the viral load (i.e., the number of virions), $L_v$, transmitted downwind to the bronchial recesses through inhalation of URT-derived particulates can be computed (with imposition of consistent length and time scale units) as:

$$L_v = \frac{144\,\pi\,d^3\,\eta\,T\,\mathcal{V}\,\mathcal{P}\,(d_i)}{\delta t} \times 10^{-12},\tag{17}$$

where $L_v$ is functionally dependent on the numerically assessed bronchial transmission rate $\eta$ (in %) and $\mathcal{P}\,(d_i)$ is the particle size density function that quantifies the number of microdroplets of size $d_i$ generated during each inhalation cycle. For simplicity, the data in Table 2 and Fig 5f enforced $\mathcal{P}$ = 1, irrespective of the particulate size, i.e., the particulate formation rate was conservatively assumed to be 1 per breathing cycle, for each test particulate size. Also, while the reported numbers in Table 2 and Fig 5f are based on the estimates guided by Eq 17, the zero viral load assessment, for example, for the 1-$\mu$m particulates implies that the corresponding transmission estimate from Eq 17 resulted in a fractional number $\ll 1$, and bears no physical significance. To note here additionally, for the assumed $\mathcal{P}$, the quantity $L_v$ exceeded 300 for $d \in [13, 22]\,\mu$m. See the conclusion section later for the related translational relevance in infection mechanics.

## Heaviside trend governing bronchial deposition and penetration

Plotting the cumulative transmission efficiencies from Table 1 (last column) with particulate diameters $d$ (in $\mu$m) along the horizontal axis reveals an inverted S-trend with (as expected) a high bronchial penetration for smaller particulate sizes, followed by a gently sloped dip; see Fig 5e. The behavior could be approximated with a modified Heaviside function [69] of the following empirical form:

$$\eta \text{ (in \%)} \approx \mathcal{C}_1 + e^{\mathcal{C}_2/\left[1+e^{\wp(d+\mathcal{C}_3)}\right]}, \text{ with } d \mapsto \eta.\tag{18}$$

In above, $\wp$ and $\mathcal{C}_i$, with $i \in \{1, 2, 3\}$, are constant fitting parameters. The adopted multivariate Heaviside formalism is particularly suitable for modeling the inverted S-shaped trend of bronchial deposition and penetration as a function of the particulate sizes, owing to its ability to represent sharp yet smooth transitions between high and low deposition rates across different size thresholds. This function effectively captures the nonlinear behavior implicit to respiratory systems, where smaller particulates penetrate deeply into the bronchial pathways, while larger ones tend to deposit more proximally or are filtered out. The mathematical *ansatz* allows for adjustable steepness and transition points, rendering it flexible enough to fit the specific inflection points associated with particle size changes. Herein, the solid red curve, fitting through the numerical data-points, was derived using the Nelder-Mead simplex algorithm for systematic error minimization [70,71]. The algorithm, also known as the downhill simplex method, is an iterative, heuristic optimization technique

designed for unconstrained optimization problems, especially where derivative information is not available or the function is noisy or discontinuous. The curve in Fig 5c invokes the following fitting parameter values:

$$\mathcal{C}_1 = -2.1706, \quad \mathcal{C}_2 = 4.6755, \quad \mathcal{C}_3 = -23.9592, \quad \wp = 0.23468. \tag{19}$$

**Representative experimental comparison of the penetration trend**

Fig 5g compares the numerically simulated penetration trend with sample published experimental data (see figure 5 in [66]), by examining the dependence of $\eta$ on the ratio of Stokes number to Reynolds number ($Stk/Re$). To generate the corresponding data from this study, the latter was calculated as:

$$\frac{Stk}{Re} = \frac{C_c\,\rho_p}{18\,\rho}\left(\frac{d}{D_{in}}\right)^2, \tag{20}$$

where $D_{in}$ denotes the hydraulic diameter at the redacted geometry's inlet (computed as $4A_{in}/P_{in}$, with $A_{in} = 118.85$ mm$^2$ being the cross-sectional area and $P_{in} = 41.64$ mm being the perimeter at the geometry inlet). The Cunningham slip correction factor was calculated as $C_c = 1 + (2.52\,\lambda_{FP}/d)$, with $\lambda_{FP} \approx 0.067$ $\mu$m representing the mean free path of air molecules surrounding the tracked particulates [24,72]. The assumption of inhaled air as a continuum would, of course, have reduced $C_c$ to 1 (with $\lambda_{FP} \approx 0$), which otherwise ranges between 1.01 and 1.17 based on the variation in $d$. The values being close to 1 do not, however, significantly impact the magnitude order of $Stk/Re$.

Based on the range of $Stk/Re$ reported in [66], $\eta$ was plotted (in Fig 5g) only for $d \in [1, 15]$ $\mu$m. The shaded region therein encapsulates the experimental data points. The graphical slopes between the experimental and numerical results are comparable. In addition, considering a deviation of $|\eta| = 2\%$ from the shaded region as indicative of disagreement with the experimental trend; 80% of the numerical data falls within the experimentally prescribed domain. The agreement stands at 100% if the permissible $|\eta|$ deviation $\to 4\%$.

**Analytical projections for particulate transport: consistent with the full-scale numerical findings**

Fig 6a demonstrates the vorticity field mapped over the section shown in Fig 4c. Notably, high vorticity regions (with magnitudes reaching $\approx 3000$ s$^{-1}$) are observed along the borders. Such localized fluid swirling can arise in intricate airway regions, impacting transport and mixing. For the present system, the local rotation of inhaled air is driven by the separating shear layers as the shape of the airway cavity widens following the laryngotracheal constriction. In the reduced-order analytical setup, the vortex patch core marked by $\mathbb{V}$ is modeled with a spatial assembly of five point vortices, as described in the methods. In Fig 6b-h, the grey curves show the streamlines of the background flow field in the S-ROAM, with the red curves tracing the sample particulate trajectories. Smaller particulates, owing to their low inertia and smaller $Stk$, follow the streamlines (on which they were embedded at entry points) more closely. For larger particulates, inertia-dominated dynamics, combined with gravitational impaction, tend to inhibit downwind penetration by biasing their motion toward deposition on the walls (see panels g and especially h, in Fig 6). However, some sufficiently large particles, such as those with diameters of 10 and 15 $\mu$m (see Fig 6c-d), can still effectively maneuver around the vortex trap to reach the lower airspace. This trend aligns with the full-scale numerical projections.

Note that the particulates entering through the middle of the inlet face of the S-ROAM are marked as Ⅰ while those entering near the right edge are marked as Ⅱ. As shown in Fig 7, the particulates Ⅱ, being nearer to the vortex region, are deviated more (from the airflow streamlines they were embedded on at the channel inlet) compared to the particulates Ⅰ. The deviation generically grows as the particulate diameters are increased, which aligns with the numerical findings reported in Table 1. A greater deviation implies that the particulates are being increasingly shifted toward the channel walls, which in turn would result in declining deposition and penetration levels in the downwind lower airway.

**Fig 6. Simulation-guided analytical modeling in the laryngotracheal space.** (a) Simulated vorticity contour map on a representative two-dimensional cross-section (the S-ROAM domain) running approximately midway through the cavity; see Fig 4b. The core of the dominant vortex patch is conservatively bounded in the rectangle, marked by $\mathbb{V}$, with its area equating the vortex patch area $A$ used in the S-ROAM. Panels (b-h) show two sample particle pathlines (in red) against the S-ROAM streamlines (in grey), respectively for particle sizes 5, 10, 15, 20, 25, 30, and 50 $\mu$m. The black lines on either side of the S-ROAM domain mark the slip wall boundaries. In each case, the vortex patch $\mathbb{V}$ is mimicked by a set of five point vortices embedded on the two-dimensional flow field with background unidirectional flow in the $\beta$-direction. The representative particles entering near the middle of the inlet face and near the right edge of the model channel are respectively marked as $\mathbb{I}$ and $\mathbb{III}$.

**Fig 7. Particle transport trend in the analytical case.** Comparison of the absolute deviation ($\tilde{\mathbb{D}}$, normalized with respect to the S-ROAM channel width) of the particle pathlines from the respective streamline they were embedded on at the inlet face. The deviation is measured along the $\alpha$-direction (see Fig 6). Deviation curves for particles entering near the middle and the right edge of the model channel inlet are respectively marked as $\mathbb{I}$ and $\mathbb{III}$.

## Discussion: Perspectives on enhancing the biophysical realism of the modeling approach

### On the interfacial mechanics at mucociliary layers during inhalation

While the numerical and analytical models presented here conclude that particulates of the length scale $\mathcal{O}(10^1)$ $\mu$m, formed through shearing and disintegration of intra-URT mucosal filaments, can indeed have significant deposition and penetration along the bronchial tubes and in the deeper lung recesses (see Table 1); to enhance the physiological realism of the model—it is essential that we obtain the actual size distribution and formation rate of the liquid particulates generated through viscoelastic separation as the inhaled air brushes past the URT mucus. The present study used the conservative estimate that 1 particulate of each size tested is formed during each breathing cycle and is a limitation.

The reader should however note that unlike expulsion regimes (during exhalation), where imaging-based data collection is relatively straightforward with human subjects expelling particulates into the outside air for different speech and breathing parameters, the current problem of characterizing the internal reverse transport (inhalation) into the lower airway in live subjects could be somewhat challenging. The approach could consequently be two-fold with synergistic numerical modeling and experimental visualizations. One can consider a 2-phase interaction on anatomically realistic upper airway surfaces; the two phases being the mimicked versions of inhaled air (phase 1) and the relatively static mucosal substrate (phase 2); see Fig 8a, and explore the interfacial mechanics leading to particulate formation and release, along with their generation rate, spatiotemporal growth, and size distribution.

### On experimental validation of deep lung penetration through in vitro physical tests

This study includes a representative validation of the model predictions against a small published dataset [66]; for details, see Fig 5g and the fourth subsection under Results. While the experimental comparison is, in essence, of *zeroth-order*, addressing this limitation through more rigorous quantitative validation of the simulated penetration and deposition profiles is potentially achievable, as outlined next. To verify the numerical modeling of the overall spatial transport, further experiments could be conducted within 3D-printed anatomically realistic airway casts internally coated with (say) concentrated glycerol (with tuned dilution levels) standing in for mucus. Controlled air flux, with embedded aerosols, could

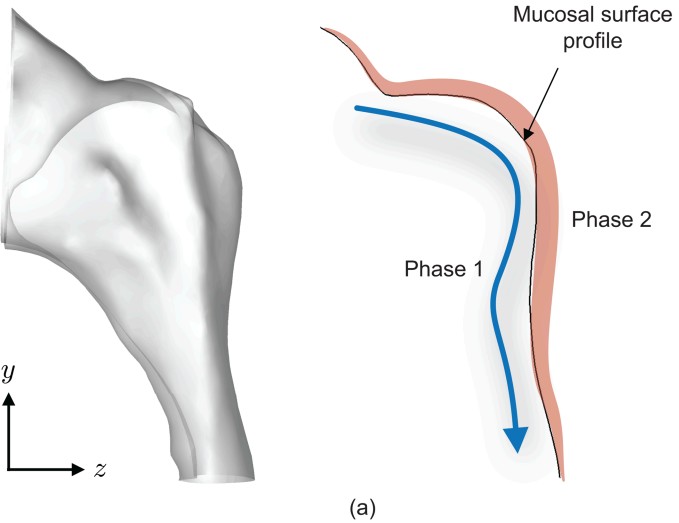
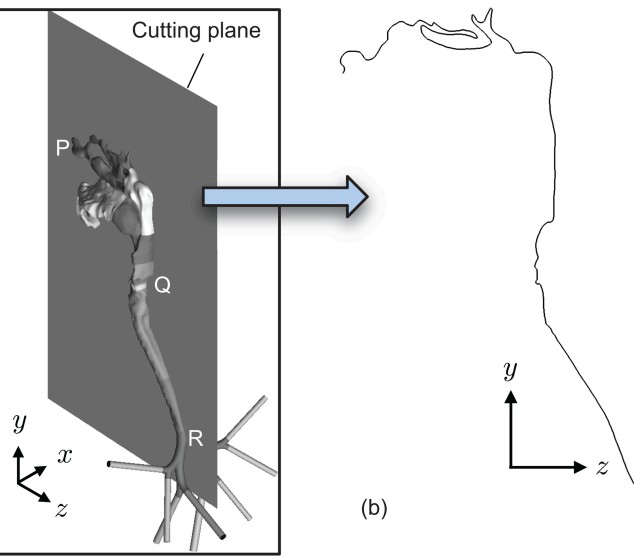

**Fig 8. Interfacial interactions and morphological complexity along mucus-coated upper airway walls.** (a) Representative planar outline of the nasopharyngeal surface topology. Phase 1 = inhaled air; Phase 2 = mucus substrate. The blue arrow indicates the inhaled air flux sweeping over the mucus (in pale red). (b) Tortuosity measurement of the anatomical space using a sagittal cutting plane.

be passed through the cast via suction mechanism set up with a vacuum pump. One approach (among others) to obtain highly resolved data on lower airway penetration could be to use the sophisticated gamma scintigraphy technique [73], wherein the solution to be aerosolized and administered into the cast (with the incoming air) would be seeded with a mildly radioactive element (e.g., Technetium). After the particulates have landed along the airway walls, the radioactive signals (emitted by the deposited mass and thus presenting a measure of local penetration) could be compared with the in silico deposition patterns along the bronchial pathways. In this context, the reader may scan our previous works with experimental validations [73,74]; in particular, [73] called on gamma scintigraphy measurements within anatomically accurate and transparent 3D-printed airway casts to physically verify numerically modeled sinonasal deposition from over-the-counter nasal sprays. The modeling protocol therein implemented similar computational schemes as in the present analysis.

### On the translational inputs for evaluating URT-to-LRT pathogen transport

The fluid dynamics findings, at their core, quantify the efficiency ($\eta$, in %) of URT-derived microdroplets in penetrating the bronchial passages and demonstrate that larger particulates can effectively reach the lower respiratory recesses if their origins are in the URT (see Table 1). To project the pathogen load ferried to the lungs by these particulates, Eq 17 connects $\eta$ with the following assumptions: (a) a fixed virion concentration in the mucosal substrate, $\mathcal{V}$, based on known data [25] for the test pathogen (SARS-CoV-2); (b) a particulate formation rate of one per breathing cycle for each tested particulate size; and (c) a constant 5-second duration ($\delta t$) for a complete breathing cycle [68]. The study also assessed URT-to-LRT virion transmission over a period of $T = 3$ days, consistent with typical durations observed for rapid onset of lung infection following URT symptoms [15,67]. These conservative assumptions provide a plausible mechanistic explanation for the accelerated development of secondary lung infections (see Fig 5f and Table 2; additionally, refer to the second section under Conclusion). To maximize the potential of the pathogen-agnostic framework presented here, future studies could expand beyond these assumptions through a broader parametric sweep, with realistic variations in $\delta t$, $T$, particle formation rates (see the first section under Discussion), and $\mathcal{V}$ (which besides depending directly on the specific pathogen type and disease prognosis, may also be a function of the particulate sizes and the corresponding fragmentation mechanism [75]).

### On limitations implicit to the modeling framework and its clinical relevance (see also Table 3)

In terms of achieving true biological realism, it is important to note the limitations implicit to the numerical modeling and theoretical analysis described in this study. The approach here algorithmizes the complex dynamics of infection onset, particularly omitting the role of immune responses and mucosal properties. In an actual infection setting, the innate and adaptive immune responses would be activated, likely leading to dynamic alterations in mucus properties that could impact intra-URT particulate formation, clearance, and the pathogen's state within these fragments. By focusing on a static view of mucosal fragmentation and particulate generation, this study currently does not account for such dynamic physiological responses that may influence the outcomes of viral transport and downwind deposition.

Another modeling limitation herein stems from the structural rigidity of the airway cavity walls in the digitized test domain and the lack of wall-adhering mucosal motion. Wall deformations (including airway compliance, cyclic expansion/contraction, bronchoconstriction or collapse/re-opening) can alter local flow patterns, velocities, residence times, and near-wall mixing, which in turn may impact particulate transport and deposition trends [76]. For instance, rapid area or direction changes would increase inertial impaction (especially important for particulates $\gtrsim 5~\mu m$), longer residence or flow recirculation can enhance gravitational settling for smaller sizes, and unsteady motion can boost convective–diffusive capture of sub-micron particulates. The net effect depends on the particulate size, airway shapes, and the specific nature and timing of the deformation. In general, this study implicitly assumes that the time scales for such deformations are longer

**Table 3. Potential directions for future research to address current limitations and enhance the described URT-to-LRT transmission framework.**

| Area | Scope for future investigations |
|---|---|
| Physiological nuances | • Experimental observation of intra-URT mucosal fragmentation mechanics during inhalation<br>• Measurement of URT-derived particulate size distribution and their formation rates, as realistic inputs to the in silico downwind transport model<br>• Assessment of evaporative and thermo-hygroscopic changes in the particulates, especially for those with longer transport time scales while moving through deeper lung regions<br>• Precise assessment of dynamic airway deformation effects on URT-to-LRT particulate transport<br>• Consideration of alveolar kinetics, in context to bronchial infection onset |
| Numerical paradigm | • Simulations within airway reconstructions encompassing higher bronchial generations with anatomical precision<br>• Consideration of multiple breathing states and faster inhalation airflow regimes<br>• Sensitivity analysis with different distributions of particulates at laryngeal inlet and varied discrete phase boundary conditions along the airspace enclosure<br>• Incorporating a wider anatomical variability in the test domains built from medical scans |
| Validation | • Comparison of bronchial penetration/deposition profiles with in vivo and in vitro physical measurements, e.g., with sophisticated techniques such as gamma scintigraphy<br>• Calibration of reduced-order analytical schemes to match three-dimensional spatiotemporal dynamics |
| Translation to other pathogens | • Enforce parametric selections specific to the pathogen type and disease prognosis, such as for measurements related to typical shed particulate sizes ($d$) and the concentration of pathogens within the particulate material ($\mathcal{V}$). The latter, besides depending on the pathogen itself, may also vary with the particulate size and the fragmentation mechanism at the mucosal substrate. |

than the time scale of aerial particulate transport within the redacted geometry. Future studies could be designed to capture the plausible wall deformation effects through sensitivity studies with compliant-walls and resulting fluid-structure interactions.

In context to other model geometric constraints and as described in the Methods section: while the primary bronchi shapes were guided by segmented CT data (with the right bronchus wider than the left), the secondary and tertiary bronchi were digitally engineered. The anatomical feasibility of the system was verified by a respiratory care specialist at the author's institution (see Acknowledgments). Still, the smooth surface design of the secondary and tertiary bronchi may be construed as a limitation to the overall physiological realism of the model. However, it is important to recognize that the central finding of this study is that larger respiratory particulates, fractured from the upper airway mucosa, can effectively reach the lower airspaces—demonstrating greater penetration compared to particulates of similar sizes inhaled from external sources. Although surface topology details (such as those in the secondary and tertiary bronchi) may indeed influence local deposition sites owing to flow-wall interactions, these nuances are unlikely to significantly affect the overall penetration of particulates into the lower respiratory tract. Also, with the redacted test geometry extending only till generation 3 bronchial branching, $\eta_d$ reflects the penetration efficiency into the further downwind branches, based on the quantifiable transport across the distal bronchiolar outlets in the geometry. Detailed alveolar kinetics (e.g., local deposition patterns therein) are, however, outside the scope of this analysis (as noted in the Introduction). This limitation can affect translational conclusions on intra-LRT infection onset.

The next notable constraint in this study concerns the absence of evaporation and thermo-hygroscopic modeling [77,78]. The inhaled airflow was assumed isothermal and heat transfer processes were neglected, implying there was no size evolution of the moving particulates owing to such effects. This simplification could bear on the accuracy of predicted transport features and deposition patterns, especially if the environmental conditions approach extreme humidity and temperature. Accordingly, future studies could be planned with a coupled heat and mass transfer framework, enabling a more precise simulation of inhaled particulate size variations. Alternatively, conducting a sensitivity analysis on relevant parameters such as the intra-airway relative humidity and temperature could help assess the evaporative / thermo-hygroscopic conditions impacting the bronchial penetration patterns of URT-derived particulates.

This work also considers only a single breathing condition—specifically, an inhaled airflow rate of 15 L/min, assuming quiet inspiration [52]. Sensitivity of the findings to alterations in flow rates and breathing patterns [79], such as deeper breaths, tachypnea, or breath-holding pauses, is left unexplored. Future investigations should attempt to examine how different flow rates and breathing waveforms might influence airflow dynamics, the resulting mucosal fragmentation at the URT, and the subsequent downwind transport of the microdroplets. Such studies can enhance the clinical relevance of the model outcomes.

The (somewhat) simplified particulate source scheme in the redacted test domain, whereby the particulates are seeded at the laryngeal inlet facet centroids (see Fig 2l), might not accurately represent the complex, physiologically realistic distribution of particulate sources, which may vary in space, size, and release mechanisms within the URT. To improve the fidelity of the discussed framework, future work should incorporate physiology-based source maps and a realistic size spectrum of particulates, emulating actual mucosal fragmentation patterns during inhalation. In addition, the current work employs canonical discrete phase boundary-condition assumptions, whereby tracked particulates are reflected at the inlet face (to rule out non-physical loss of particulates against the streamwise direction), escape through the redacted geometry's distal bronchiolar outlets (to assess deep lung penetration), and are trapped if they approach the geometry walls (to quantify local deposition). These assumptions could be limiting by introducing biases in the spatial dynamics and deposition/penetration predictions. To address, future efforts should attempt a boundary-condition sensitivity analysis to evaluate how different assumptions therein would influence the results.

Moving on to the analytical validation, the two-dimensional S-ROAM projects the vortex effects from the three-dimensional LES data on to a mid-lying section; the location of this section is illustrated in Fig 4b-c. The goal of the analytical exercise was to verify whether the numerically simulated particulate dispersion trends are qualitatively consistent with those predicted by an idealized, reduced-order mathematical model—which they were (see Figs 6 and 7). In general, there ought to be calibration differences between the computational and analytical approaches, which do introduce errors in the two-dimensional model's particle transport analyses. Future work in this direction should involve a comprehensive calibration of the simplified model against full-scale computational results, with quantification of errors and uncertainty bands to enhance its robustness and predictive reliability.

Finally, the reported results, while establishing the basic plausibility of the URT-to-LRT transmission mechanism through the use of a test case, are nonetheless for a single CT-based anatomical geometry with Weibel-engineered distal bifurcations truncated at generation 3 of bronchial branching and with baseline parameters that do not (yet) explicitly account for patient-to-patient variability. Realistic differences in airway anatomy, mucosal characteristics, and immune response between individuals are factors that could lead to different fragmentation and transport dynamics. This limitation can have a bearing on the generalizability of the findings. Future work should incorporate sensitivity studies that test the robustness of the modeling approach across a range of physiological conditions and individual variability.

Moreover, the "so what" question remains. The findings, while *precise* in a computational sense and confirming the mechanistic possibility of rapid lung invasion orchestrated by URT-derived pathogen-laden particulates, do not yet translate into actionable clinical recommendations. Future research should assess whether the insights gained could inform clinical practices or support diagnostic tools, possibly through personalized respiratory physics models that account for specific physiological traits. Such models could potentially identify subjects at higher risk of clinical morbidity, e.g., from brisk onset of severe lung infection.

## Conclusion: The main takeaways

### Can large particulates, generated from the intra-URT mucus coating during inhalation, penetrate to the bronchial airspace?

The reported bronchial transmission trends, representatively validated against a small sample of published experimental data [66] and characterized by $\eta$ (as a function of the test inhaled particulate sizes) in the numerical experiments (see

Table 1 and Fig 5e), are found to be consistent with the first-principles reduced-order analytical findings that modeled the impact of dominant intra-airway vortex instabilities in the laryngotracheal domain on local particle transport (Fig 6). Not only the aerosols (i.e., particulates with diameters $\lesssim 5\ \mu m$) but also droplets as large as 10 and 15 $\mu m$ exhibit remarkable efficiency at reaching the bronchial spaces and deep lungs, so long as they are sheared away from the URT surface and are still airborne as they enter the larynx. This contradicts the general perception that only particulates smaller than 5 $\mu m$ can comfortably penetrate the lower airway. Missing the nuance therein is the fact that the incumbent perspective is based on the spatiotemporal mechanics of particulates inhaled from the outside air (which are thus being forced to navigate the sharp curvatures inside the anterior nasal space), and our understanding of intra-URT mucosal breakup during inhalation and the subsequent aerial advection in the downwind tract is still nascent.

For a physics-based rationale to explain the derived deposition and penetration profiles, let us consider the tortuosity $\mathcal{T}$ of the pathways to be traversed by a particulate inhaled from outside (called hereafter $P_o$), compared to a particulate that is generated within the URT and is still airborne in the laryngeal airspace (hereafter called $P_u$). Fig 8b illustrates a representative vertical cutting plane used to extract the tortuosity measurements. For $P_o$, the corresponding tortuosity is the ratio of the curved path length and the linear distance in space between the points P and R; let us represent it as $\mathcal{T}_{P_o}$. For $P_u$, it is similarly the ratio of the curved path length and the linear distance in space between the points Q and R; let us represent it as $\mathcal{T}_{P_u}$. The geometric measurements return: $\mathcal{T}_{P_o} \approx 1.90$ and $\mathcal{T}_{P_u} \approx 1.04$. Consequently, if the mechanics of $P_o$ is inertia-dominated (true for particulates $\gtrsim 5\ \mu m$), they exhibit less success at navigating the highly tortuous pathway and are deposited along the anterior URT; see Fig 3 in this context. On the contrary, particulates of similar sizes, if they only have to traverse the Q-R tract, the less tortuous pathway ensures that a high percentage of them would end up reaching the bronchial domains; see Fig 5e. As an aside, also note that the tortuosity estimates obtained here match exactly with our previously published data on mammalian airway morphology [80]; see panel (d) in figure 1 of the cited paper. Thus, the findings of this study satisfactorily address the question $Q_1$ (see Introduction).

### How do the transmitted viral loads compare to the infectious dose of the test pathogen?

Infectious dose, $I_D$, of a virus quantifies the minimum number of virions that can potentially launch infection in an exposed subject [56,81] and is a fundamental virological parameter. Independent studies by us [26,29] and others [27,82] have verified that $I_D \approx 300$, for SARS-CoV-2. Thus, evidently (per Fig 5f and Table 2), the viral load $L_v$ transmitted by, for instance, the 15-$\mu m$ droplets would alone exceed the $I_D$ threshold, thereby providing a mechanics-based rationale for the fast disease progression to the lower airway. The brisk pace is otherwise difficult to explain based exclusively on tissue level proliferation and direct deep lung inhalation of dominantly sub 5-$\mu m$ particulates from outside. However, from a translational perspective, it is critical to note that the present in silico framework does not *yet* take into account the host innate and adaptive immune responses to the invading virions which can lead to shifts in mucus properties altering breakup, while affecting the dynamics of clearance and state of the virus in the URT-derived fragments. The immunological considerations, once incorporated into the mechanics paradigm (prospectively in the form of a "correction factor" to Eq 17 while evaluating the *potent* viral load transmitted downwind), can help rationalize the varying rates of clinical prognosis recorded in different subjects [67]; e.g., deep lung infection for SARS-CoV-2 has historically ensued over a range of $2-8$ days following the appearance of initial URT symptoms. In summary, the findings do satisfactorily address the question $Q_2$ posed in the Introduction.

### Acknowledgments

The author thanks Dr. Julia Kimbell (School of Medicine, University of North Carolina Chapel Hill) for sharing existing, de-identified imaging data and Ms. Abby Wortman (Clinical and Lab Instructor for the Respiratory Care program at the

author's institution) for verifying the anatomical realism in the CT reconstruction and the scientific nomenclature on airway physiology. The author also thanks Dr. Neelesh Patankar (Northwestern University) for insightful discussions on deep lung infections.

## Author contributions

**Conceptualization:** Saikat Basu.

**Data curation:** Saikat Basu.

**Formal analysis:** Saikat Basu.

**Funding acquisition:** Saikat Basu.

**Investigation:** Saikat Basu.

**Methodology:** Saikat Basu.

**Project administration:** Saikat Basu.

**Resources:** Saikat Basu.

**Software:** Saikat Basu.

**Supervision:** Saikat Basu.

**Validation:** Saikat Basu.

**Visualization:** Saikat Basu.

**Writing – original draft:** Saikat Basu.

**Writing – review & editing:** Saikat Basu.

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
