## [Decision Letter · Decision Letter 0]

23 Jun 2025

PONE-D-25-20475On the mechanics of inhaled bronchial transmission of pathogenic microdroplets generated from the upper respiratory tract, with implications for downwind infection onsetPLOS ONE

Dear Dr. Basu,

Thank you for submitting your manuscript to PLOS ONE. After careful consideration, we feel that it has merit but does not fully meet PLOS ONE’s publication criteria as it currently stands. Therefore, we invite you to submit a revised version of the manuscript that addresses the points raised during the review process.

We look forward to receiving your revised manuscript.

Kind regards,

Saidul Islam, Ph.D.

Academic Editor

PLOS ONE

Journal Requirements:

“National Science Foundation CAREER Award (Grant Number 2339001; Fluid Dynamics program with Dr. Ron Joslin as program manager)”

4. Thank you for uploading your study's underlying data set. Unfortunately, the repository you have noted in your Data Availability statement does not qualify as an acceptable data repository according to PLOS's standards.

Reviewers' comments:

Reviewer's Responses to Questions

**Comments to the Author**

1. Is the manuscript technically sound, and do the data support the conclusions?

Reviewer #1: Yes

Reviewer #2: Yes

2. Has the statistical analysis been performed appropriately and rigorously?

Reviewer #1: Yes

Reviewer #2: Yes

3. Have the authors made all data underlying the findings in their manuscript fully available?

Reviewer #1: No

Reviewer #2: Yes

4. Is the manuscript presented in an intelligible fashion and written in standard English?

Reviewer #1: Yes

Reviewer #2: Yes

5. Review Comments to the Author

Reviewer #1: General Summary:

Saikat presented The full-scale numerical transmission trends are consistent with findings from our reduced-order mathematical model that conceptualizes the influence of intra-airway vortex instabilities on local particle transport through point vortex idealization in an anatomy-guided two-dimensional potential flow domain. The results collectively demonstrate markedly elevated lower airway penetration by URT-derived particulates, even by those as large as 10 and 15 µm. The high viral load, often exceeding the pathogen-specific infectious dose, carried by such droplets into the bronchial spaces of the sample airway provides a plausible mechanistic explanation for the accelerated seeding of secondary lung infection.

Major Points:

1. Abstract: The author mentions the existence of viscoelastic behavior but does not mention how to model it in the manuscript. Please see the following article: https://doi.org/10.1177/02676591221128141

2. Line 144: In equation 4, there is no need to write the density term, given that the flow is incompressible.

3. Due to the lack of a Nomenclature, it is almost impossible to check the equations listed exactly.

4. Line 171: Have you obtained the time steps of 0.0002 s based on the Courant number?

5. Line 211: Have you considered 1622 particulates based on the number of inlet compute nodes?

6. Since you did not consider the deformation of the wall, please mention this as a computational limitation in the discussion. Does this limitation not affect the response? Please see the following article: https://doi.org/10.1016/B978-0-12-823913-1.00009-9

7. Add more details about mesh quality, boundary layer height, Y-plus quantity, etc.

8. In Figure 6, are the Vorticity changes up to 3000 reasonable on the border?! Please explain.

9. Please present the ethics approval document and explain how much radiation dose and in how many layers you were able to obtain the lower airway generations.

10. Line 167: Specify the numerical value of the pressure inlet and outlet boundary condition.

Reviewer #2: This study investigates the role of microdroplets derived from mucosal liquids in the upper respiratory tract and their contribution to the rapid onset of deep lung infections following upper respiratory tract (URT) infections. Utilizing three-dimensional airway reconstructions and airflow simulations. While this is a robust study with well-discussed results, several issues must be addressed prior to publication in PLOS ONE:

1. It appears that the primary, secondary, and tertiary bronchi were not extracted from CT scan images, which may influence the particle deposition fraction in these regions.

2. The findings related to deposition efficiency in the trachea and various bronchi should be compared with previous studies involving inhaled particles to highlight any differences.

Minor Comments:

1. In the numerical section, providing more detailed information regarding the generation of grid meshes and the numerical equations solved on these grids would be beneficial. Additionally, a more comprehensive explanation of the numerical methods employed would enhance clarity.

2. The velocity and particle boundary conditions need to be specified for various regions to ensure accuracy.

3. In Figures 3 and 5, the y-axis of the contour plots appears to be of limited utility. It may be more effective to present this data in an x-y plot, with the x-axis representing diameter and the y-axis representing η or ηd, as this would better illustrate the trends, similar to Figure 5c. Furthermore, it would be preferable to report the results for each generation separately.

4. The results presented in Table 1 should also be delineated for each generation individually.

Addressing these revisions would significantly enhance the clarity and impact of the study.

6. PLOS authors have the option to publish the peer review history of their article (what does this mean?). If published, this will include your full peer review and any attached files.

Reviewer #1: **Yes: **Hamidreza Mortazavy Beni

Reviewer #2: No

---

## [Author Response · Author response to Decision Letter 1]

28 Aug 2025

I sincerely appreciate your inputs for this manuscript. The following key files are included in the revision package:

1. Detailed point-by-point response letter addressing the review comments

2. Marked-up version of the manuscript with track changes visible

3. Clean, compiled version of the revised manuscript

4. Revised figure files

In the marked-up version, newly added material is highlighted in royal blue, while deletions are shown in red strike-through.

I hope the revised manuscript meets your approval. Once again, I sincerely thank the reviewers for their constructive critique and valuable suggestions.

Sincerely,

Saikat Basu, Ph.D.

Associate Professor of Mechanical Engineering

South Dakota State University

---

## [Decision Letter · Decision Letter 1]

7 Oct 2025

PONE-D-25-20475R1On the mechanics of inhaled bronchial transmission of pathogenic microdroplets generated from the upper respiratory tract, with implications for downwind infection onsetPLOS ONE

Dear Dr. Basu,

Thank you for submitting your manuscript to PLOS ONE. After careful consideration, we feel that it has merit but does not fully meet PLOS ONE’s publication criteria as it currently stands. Therefore, we invite you to submit a revised version of the manuscript that addresses the points raised during the review process.

We look forward to receiving your revised manuscript.

Kind regards,

Saidul Islam, Ph.D.

Academic Editor

PLOS ONE

Journal Requirements:

Reviewer's Responses to Questions

**Comments to the Author**

1. If the authors have adequately addressed your comments raised in a previous round of review and you feel that this manuscript is now acceptable for publication, you may indicate that here to bypass the “Comments to the Author” section, enter your conflict of interest statement in the “Confidential to Editor” section, and submit your "Accept" recommendation.

Reviewer #1: (No Response)

Reviewer #2: (No Response)

2. Is the manuscript technically sound, and do the data support the conclusions?

Reviewer #1: (No Response)

Reviewer #2: (No Response)

3. Has the statistical analysis been performed appropriately and rigorously?

Reviewer #1: (No Response)

Reviewer #2: (No Response)

4. Have the authors made all data underlying the findings in their manuscript fully available?

Reviewer #1: (No Response)

Reviewer #2: Yes

5. Is the manuscript presented in an intelligible fashion and written in standard English?

Reviewer #1: (No Response)

Reviewer #2: Yes

6. Review Comments to the Author

Reviewer #1: Minor Revision:

For any item you cannot fully address now, state it explicitly as a study limitation in the Discussion.

1. Insufficient experimental validation — Current text mentions only “representative” validation against a small published set; add quantitative comparisons (e.g., gamma-scintigraphy or 3D-printed cast tests) for penetration/deposition profiles. If not feasible, flag as a limitation.

2. No evaporation / thermo-hygroscopic growth modeling — The flow is isothermal and heat transfer is discounted, so droplet size evolution is ignored; add a coupled heat/mass-transfer model or at least a sensitivity analysis to RH/temperature. If not feasible, note as a limitation.

3. Single anatomy, truncated at G3 — One subject (CT-based) with Weibel-engineered distal branches; include ≥2 additional anatomies and/or extend generations, or state generalizability limits.

4. Rigid walls / no FSI or mucosal motion — Add a compliant-wall (or FSI) sensitivity to capture deformation effects; otherwise declare as a modeling limitation.

5. Single breathing condition (15 L/min, quiet inspiration) — Report sensitivity to flow rate and breathing waveforms (deeper breaths, tachypnea, pauses). If not feasible, mark as a limitation.

6. Simplified particle source model — Particles are seeded uniformly at inlet facet centroids; replace with physiologic source maps and realistic size spectra, or state as a limitation.

7. DPM boundary-condition assumptions — Using reflect (inlet), escape (outlets), trap (walls) can bias trajectories; add a boundary-condition sensitivity check. If not possible, list as a limitation.

8. Reduced-order 2D model (S-ROAM) calibration — The point-vortex 2D idealization needs clearer calibration vs 3D LES (errors/uncertainty bands); otherwise state its scope as limited/illustrative.

9. Viral-load projection assumptions — Fixed sputum concentration, “one particle per breath per size,” and T=3 days are strong assumptions; add parameter sweeps (V, generation rate, δt, T). If not feasible, declare as a limitation.

10. Scope ends at G3 (no alveolar deposition) — Clarify that alveolar kinetics are out of scope; if not extending, state this explicitly as a limitation affecting translational conclusions.

Reviewer #2: The author provided a detailed explanation of my comments; however, I have one concern regarding the discussion of velocity and particle boundary conditions within the geometry. The author did not adequately address this in the revised manuscript.

Specifically, it is essential to specify the type of boundary conditions at the inlet, such as a parabolic velocity profile for the velocity and a uniform particle distribution for the particles, as well as to provide similar details for the outlet. Merely naming the reflect and escape boundaries in a numerical paper is insufficient and does not meet the expected standards of clarity and rigor.

The paper can be accepted contingent upon this minor revision being made.

7. PLOS authors have the option to publish the peer review history of their article (what does this mean?). If published, this will include your full peer review and any attached files.

Reviewer #1: **Yes: **Hamidreza Mortazavy Beni

Reviewer #2: **Yes: **MohammadHadi Sedaghat

---

## [Author Response · Author response to Decision Letter 2]

16 Oct 2025

I sincerely appreciate your inputs for this manuscript. In this revision, I have addressed all the suggestions from the 2nd round of peer review. The following key files are included in the revision package:

1. Detailed point-by-point response letter addressing the review comments

2. Marked-up version of manuscript with track changes (green for latest Round 2 revisions; blue for prior Round 1 revisions)

3. Clean, compiled version of the revised manuscript

4. All figure files

I hope the revised manuscript meets your approval. Your constructive comments have been instrumental in enhancing the quality of this paper.

Sincerely,

Saikat Basu, Ph.D.

Associate Professor of Mechanical Engineering

South Dakota State University

---

## [Editor Report · Decision Letter 2]

20 Oct 2025

On the mechanics of inhaled bronchial transmission of pathogenic microdroplets generated from the upper respiratory tract, with implications for downwind infection onset

PONE-D-25-20475R2

Dear Dr. Basu,

We’re pleased to inform you that your manuscript has been judged scientifically suitable for publication and will be formally accepted for publication once it meets all outstanding technical requirements.

Kind regards,

Saidul Islam, Ph.D.

Academic Editor

PLOS ONE
---

## [Editor Report · Acceptance letter]

PONE-D-25-20475R2

PLOS ONE

Dear Dr. Basu,

I'm pleased to inform you that your manuscript has been deemed suitable for publication in PLOS ONE. Congratulations! Your manuscript is now being handed over to our production team.

Kind regards,

on behalf of

Dr Saidul Islam

Academic Editor

PLOS ONE